# NAAA-regulated lipid signaling in monocytes controls the induction of hyperalgesic priming in mice

Yannick Fotio [1], Alex Mabou Tagne [1,10], Erica Squire[1,10], Hye-lim Lee[1], Connor M. Phillips[2], Kayla Chang [1], Faizy Ahmed[1], Andrew S. Greenberg [3], S. Armando Villalta[2,4], Vanessa M. Scarfone[5], Gilberto Spadoni[6], Marco Mor [7] & Daniele Piomelli [1,8,9] ✉

Circulating monocytes participate in pain chronification but the molecular events that cause their deployment are unclear. Using a mouse model of hyperalgesic priming (HP), we show that monocytes enable progression to pain chronicity through a mechanism that requires transient activation of the hydrolase, *N*-acylethanolamine acid amidase (NAAA), and the consequent suppression of NAAA-regulated lipid signaling at peroxisome proliferator-activated receptor-α (PPAR-α). Inhibiting NAAA in the 72 hours following administration of a priming stimulus prevented HP. This effect was phenocopied by NAAA deletion and depended on PPAR-α recruitment. Mice lacking NAAA in CD11b⁺ cells – monocytes, macrophages, and neutrophils – were resistant to HP induction. Conversely, mice overexpressing NAAA or lacking PPAR-α in the same cells were constitutively primed. Depletion of monocytes, but not resident macrophages, generated mice that were refractory to HP. The results identify NAAA-regulated signaling in monocytes as a control node in the induction of HP and, potentially, the transition to pain chronicity.

Chronic pain places an enormous burden on hundreds of millions of people worldwide[1–3], yet its mechanistic underpinnings remain largely unknown[4,5]. One major challenge is the identification of molecular events that allow acute pain episodes – which typically accompany self-resolving, localized tissue injuries – to progress into persistent painful states that outlast the initial damage and can radiate outside its tissue boundaries[4]. In addition to neuroplastic changes, whose roles are well established[6,7], activation of the innate immune system has emerged as a driving factor in the progression to pain chronicity[8–11]. Studies in mice have shown, for example, that blood-borne monocytes infiltrate the spinal cord[12] and act in synergy with local microglia to promote pain chronification after nerve injury[13,14]. Similarly, monocytes and macrophages that express the CX3CR1 receptor contribute to arthritis pain and chemotherapy-induced allodynia by interacting with nociceptive neurons in dorsal root ganglia (DRG)[15,16]. Despite these advances, the molecular checkpoints that trigger the deployment of monocyte-derived cells during the transition from acute to chronic pain are still poorly understood[8–10].

An experimental model developed to investigate this transition exploits a form of neuroplasticity in which the administration of a

[1]Department of Anatomy and Neurobiology, University of California Irvine, Irvine, CA, USA. [2]Department of Physiology and Biophysics, University of California Irvine, Irvine, CA, USA. [3]Human Nutrition Research Center, Tufts University, Boston, MA, USA. [4]Department of Neurology, University of California Irvine, Irvine, CA, USA. [5]Sue and Bill Gross Stem Cell Research Center, University of California Irvine, Irvine, CA, USA. [6]Dipartimento di Scienze Biomolecolari, Università di Urbino "Carlo Bo,", Urbino, Italy. [7]Dipartimento di Scienze degli Alimenti e del Farmaco, Università di Parma, Parma, Italy. [8]Department of Biological Chemistry, University of California Irvine, Irvine, CA, USA. [9]Department of Pharmaceutical Sciences, University of California Irvine, Irvine, CA, USA. [10]These authors contributed equally: Alex Mabou Tagne, Erica Squire. ✉e-mail: piomelli@hs.uci.edu

proinflammatory substance sensitizes a subset of peripheral nociceptors to subsequent noxious and non-noxious stimuli[17,18]. In the report that first described this phenomenon[19], termed 'nociceptive' or 'hyperalgesic priming' (HP), Levine and collaborators showed that the inflammatory reaction produced by intraplantar injection of carrageenan in rats was followed by a prolonged (≥3 weeks) enhancement of the nocifensive response to other proalgesic agents applied at the same site. Further studies by the Levine laboratory identified non-peptidergic isolectin B4[+] nociceptors as cellular substrates for HP and showed that its emergence required the recruitment of protein kinase Cε-dependent signal transduction pathways[19–21]. In addition to its clinical relevance[18], HP has three notable advantages as a model to explore the progression to pain chronicity. First, it can be induced by administering a single inflammatory agent with identified mechanism of action – e.g., interleukin 6 (IL-6) or tumor-necrosis factor-α (TNF- α) – thus allowing control over a key experimental variable. Second, the initial nocifensive reaction to a priming stimulus can be suppressed by analgesic drugs with no consequence on HP induction, a clear indication that the two processes are temporally overlapping but mechanistically different[22,23]. Finally, HP unfolds with a distinctive time-course in which the immediate response to the priming trigger is separated from the appearance of the primed state by a 72 h-long interval[23,24]. This delay, which has been attributed to axonal and nuclear processes[23,25], identifies three phases of HP, which can be studied in relative isolation: 'initiation', which occurs in parallel with the inflammatory reaction to the priming trigger; 'incubation', which follows the resolution of such reaction and precedes the appearance of the primed state; and 'maintenance', which allows such state to persist for several weeks.

In the present study, we used a combination of genetic and pharmacological strategies to determine whether the intracellular cysteine hydrolase, *N*-acylethanolamine acid amidase (NAAA)[26], contributes to the emergence of HP. NAAA catalyzes the hydrolytic deactivation of palmitoylethanolamide (PEA) and oleoylethanolamide (OEA), two lipid-derived agonists of the ligand-operated transcription factor, peroxisome proliferator-activated receptor-α (PPAR-α)[27,28]. In addition to its well-established roles in the modulation of acute nociception[29] and inflammation[30], NAAA-regulated lipid signaling at PPAR-α has been implicated in the induction of persistent pathological nociception by chemical or surgical damage to peripheral tissues[31]. Building on those findings, we show now that circulating monocytes drive the progression to pain chronicity through a cell-autonomous mechanism that requires the suppression of intracellular NAAA-regulated PPAR-α signaling during – but not before or after – the incubation phase of HP. The results identify NAAA in monocytes as a control node in the development of HP and a possible target to prevent chronic pain.

## Results

### NAAA is required for HP initiation
Following an established protocol[19,32], we evoked HP in male mice by intraplantar (i.pl.) injection of IL-6 (0.5 ng) (Fig. 1A). As expected, the cytokine produced a state of ipsilateral heat and mechanical hypersensitivity that lasted <24 h and was followed by a persistent enhancement of the nocifensive response to the proalgesic eicosanoid PGE_2 (100 ng, i.pl., administered 6 days later in the same paw) (Fig. S2). The initial hypersensitivity was attenuated when mice were given, 2 h before IL-6, either the selective NAAA inhibitor ARN19702 (30 mg-kg$^{-1}$, IP)[33,34] (Fig. 1B) or gabapentin (50 mg-kg$^{-1}$, IP) (Fig. 1D). Despite these acute antinociceptive effects, the two agents did not prevent HP initiation, as shown by their inability to normalize the response to PGE_2 in IL-6-primed mice (Fig. 1C, E). The results, which were replicated in female animals (Fig. S3A–C), confirm prior studies indicating that the inflammatory reaction to a priming stimulus and the initiation of HP overlap in time but are governed by distinct mechanisms[22,23].

As the initiation and maintenance phases of HP are separated by an incubation period of ~72 h[23,24], we asked whether administering ARN19702 or gabapentin during this time interval might affect HP progression. We gave once-daily injections of ARN19702 (30 mg-kg$^{-1}$, IP), gabapentin (50 mg-kg$^{-1}$, IP) or their vehicles to male and female mice on the first, second, and third day after IL-6 (Fig. 1A). When tested using this protocol, ARN19702 blocked HP initiation (Fig. 1F, G, data from female mice are shown in Fig. S3F, G; data on mechanical hypersensitivity in male mice are shown in Fig. S4) whereas gabapentin did not (Fig. S5). The same ARN19702 regimen was also protective in male mice challenged with three additional priming interventions: TNF-α (100 ng, i.pl.) (Fig. S6A–C), carrageenan (1%, i.pl.) (Fig. S6D, E), and surgical paw incision (Fig. S7). Underscoring the strict time-dependence of its effects, ARN19702 failed to prevent IL-6-induced HP when administered in male mice for shorter time intervals (e.g., only on days 1 and 2 after IL-6) or after the end of the incubation period (e.g., on days 4 and 5 after IL-6) (Fig. 1H–J).

To further probe NAAA's role in priming induction, we studied the response to IL-6 in male mice constitutively lacking the enzyme. Previous studies have shown that inflammatory and acute nociceptive responses are blunted in homozygous *Naaa$^{-/-}$* mice[31,35]. Consistent with those findings, the mutants' acute reaction to IL-6 was weaker than that of their wild-type littermates (Fig. 1K). *Naaa$^{-/-}$* mice exhibited a normal acute response to carrageenan, however, possibly owing to the greater intensity of this stimulus (Fig. S4F). Furthermore, as seen with ARN19702 treatment, *Naaa$^{-/-}$* mice failed to develop HP after injection of IL-6 (Fig. 1L) or carrageenan (Fig. S4G). Heterozygous *Naaa$^{+/-}$* mice also showed a diminished inflammatory reaction to IL-6 (Fig. 1K) but not to carrageenan (Fig. S4G) and were unable to progress to HP after administration of either stimulus (Fig. 1L). The findings indicate that inflammatory agents induce HP through a mechanism that requires NAAA activity. Such requirement is *(i)* independent of NAAA's facilitatory role in the nociceptive response that accompanies inflammation; *(ii)* restricted to the 72-h incubation period that precedes the emergence of HP; and *(iii)* sexually monomorphic.

### NAAA initiates HP by suppressing PPAR-α signaling
Activation of the nuclear receptor PPAR-α by the endogenous NAAA substrates, PEA and OEA, underpins the anti-inflammatory and antinociceptive properties of NAAA inhibitors[34,35] as well as the agents' ability to stop the transition to pain chronicity after sciatic nerve ligation or i.pl. formalin injection[31]. To determine whether a similar mechanism might be involved in halting HP induction, we challenged male mice with IL-6 (0.5 ng, i.pl.) and treated them once daily for the subsequent three days with either ARN19702 (30 mg-kg$^{-1}$, IP) or a combination of ARN19702 and a maximally effective dose of the selective PPAR-α antagonist GW6471 (4 mg-kg$^{-1}$, IP)[31,34,36]. When administered alone, ARN19702 prevented HP induction, and this effect was abolished by the addition of GW6471 (Fig. 2A). By contrast, co-administration of ARN19702 with maximally effective doses of selective antagonists for PPAR-γ (T0070907, 1 mg-kg$^{-1}$, IP)[37] or CB_2 cannabinoid receptors (AM630, 3 mg-kg$^{-1}$, IP)[38], which have been implicated in some actions of PEA[39,40], did not substantially alter the response to ARN19702 (Fig. 2B, C). Further support for a role of PPAR-α in mediating such response was provided by the finding that PEA (30 mg-kg$^{-1}$, subcutaneous) and the synthetic PPAR-α agonist GW7647 (10 mg-kg$^{-1}$, IP) prevented priming when administered on days 1-3 post-IL-6 (Fig. 2D, E). The results suggest that inhibiting NAAA activity during the incubation phase of HP stops the progression to pain chronicity by heightening PPAR-α activation.

### Peripheral, not central, NAAA initiates HP
NAAA is expressed in CNS neurons and microglia[26,41,42]. To assess whether the development of HP might involve a central pool of the enzyme, we utilized the β-lactam-based NAAA inhibitor ARN726

(Fig. 2F), which is potent and selective but has limited CNS activity[43]. Fifteen minutes after systemic administration, ARN726 (10 mg-kg$^{-1}$, IP) reached peak concentrations of 333.8 ng-mL$^{-1}$ in plasma and 42.7 ng-mg$^{-1}$ in spinal cord (Fig. 2F, other PK properties of ARN726 are reported in Table S2). The tissue-to-plasma ratio calculated from these concentrations (=0.13) is ~56% lower than the brain-to-plasma ratio achieved by ARN19702 after oral administration of a 3 mg-kg$^{-1}$ dose (=0.30)[33]. Concomitant with its plasma peak, ARN726 caused a transitory but substantial elevation of PEA and OEA levels in circulation (Fig. 2G, H) but had no such effect in the spinal cord (Fig. 2J, K). Moreover, confirming its selectivity for NAAA[43], ARN726 did not affect the circulating levels of anandamide (Fig. 2I, L), an endocannabinoid substance that is structurally related to PEA and OEA but is primarily hydrolyzed by fatty acid amide hydrolase[26]. Despite this lack of CNS activity, systemic administration of ARN726 (1-30 mg-kg$^{-1}$, IP) on days 1-3 after IL-6 blocked the development of priming with considerable

potency (ED$_{50}$ = 2.2 mg-kg$^{-1}$) (Fig. 2M, Fig. S8). By contrast, no such effect was observed when ARN726 (30 ng) was delivered directly to the spinal cord via intrathecal infusion (Fig. 2N). We conclude that HP initiation requires NAAA activity in peripheral tissues.

## NAAA in CD11b$^+$ myeloid cells initiates HP

Outside the CNS, NAAA is predominantly (albeit not exclusively) expressed in monocytes, macrophages, and B-lymphocytes[43,44]. Because monocytes and macrophages have been implicated in the progression to pain chronicity[12,13], we examined whether NAAA-regulated lipid signaling in this cell population might contribute to HP. We generated mutant mice that selectively lack NAAA in CD11b$^+$ cells (Fig. 3A), which include monocytes, macrophages, and neutrophils[45,46]. Successful recombination was confirmed by subjecting blood samples to fluorescence-activated cell sorting (FACS) coupled to RT-qPCR analysis (Fig. 3B, C). Naaa$^{CD11b-/-}$ mice were viable,

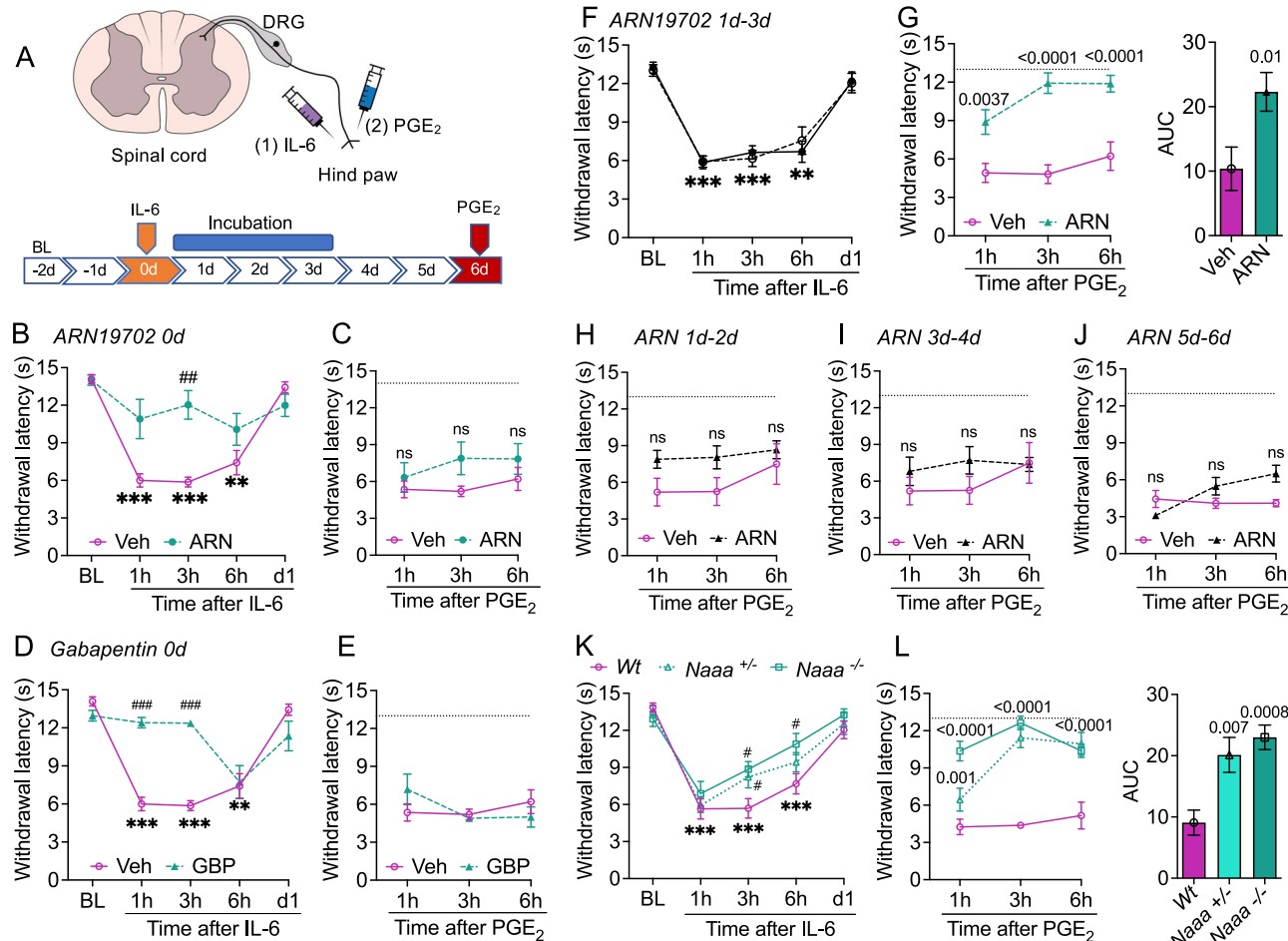

**Fig. 1 | NAAA is required to induce hyperalgesic priming. A** Model used in the present study. In most experiments, priming was induced in mice by intraplantar administration of IL-6 and was assessed 6 days later by injecting PGE$_2$ at the same site. Ipsilateral heat hypersensitivity (paw withdrawal latency, seconds) was measured under baseline (BL) conditions (−2d, −1d) and immediately after administration of IL-6 (0d), PGE$_2$ (6d), or vehicle (0d and 6d). The blue bar marks the incubation period for priming. The illustration was generated, in part, with BioRender.com. **B, D** Self-resolving heat hypersensitivity elicited by IL-6 in mice that had received 2 h earlier (**B**) ARN19702 (ARN, 30 mg-kg$^{-1}$; green circles; $n = 8$), (**D**) gabapentin (GBP, 50 mg-kg$^{-1}$; green triangles; $n = 8$), or their vehicles (Veh, magenta circles; $n = 8$). **C, E** Effects of PGE$_2$ in IL-6-primed mice treated with ARN19702 (**C**), gabapentin (**E**), or their vehicles. **F, G** IL-6-primed mice were treated on 1d-3d with vehicle ($n = 16$) or ARN19702 (30 mg-kg$^{-1}$; $n = 17$). **F** Response to IL-6 before administration of vehicle (open symbols) or ARN19702

(closed symbols). **G** Effects of PGE$_2$ in IL-6-primed mice that had received vehicle (magenta circles) or ARN19702 (green triangles). Right panel, area under the curve (AUC). **H–J** IL-6-primed mice ($n = 10$ per group) were treated with vehicle (magenta circles) or ARN19702 (30 mg-kg$^{-1}$, filled triangles) for 2 days on 1d-2d, 3d-4d, or 5d-6d, and the response to PGE$_2$ was assessed 3 days later. ARN19702 on (**H**) 1d-2d, (**I**) 3d-4d, and (**J**) 5d-6d. (**K, L**) Effects of (**K**) IL-6 (0d) and (**L**) PGE$_2$ (6d) in wild-type (Wt, magenta circles, $n = 10$), heterozygous Naaa$^{+/-}$ (green triangles, $n = 11$), and homozygous Naaa$^{-/-}$ mice (green squares, $n = 11$). Data are presented as mean ± S.E.M. and were analyzed by two-way repeated measure ANOVA followed by Bonferroni's multiple comparison test, when necessary. ***$P < 0.001$, **$P < 0.01$, *$P < 0.05$, versus baseline; ###$P < 0.001$, ##$P < 0.01$, #$P < 0.05$, versus vehicle or Wt. ns, non-significant. $P$ values versus vehicle are shown, when possible. Dotted lines indicate baseline withdrawal latency. Source data are provided as a Source Data file.

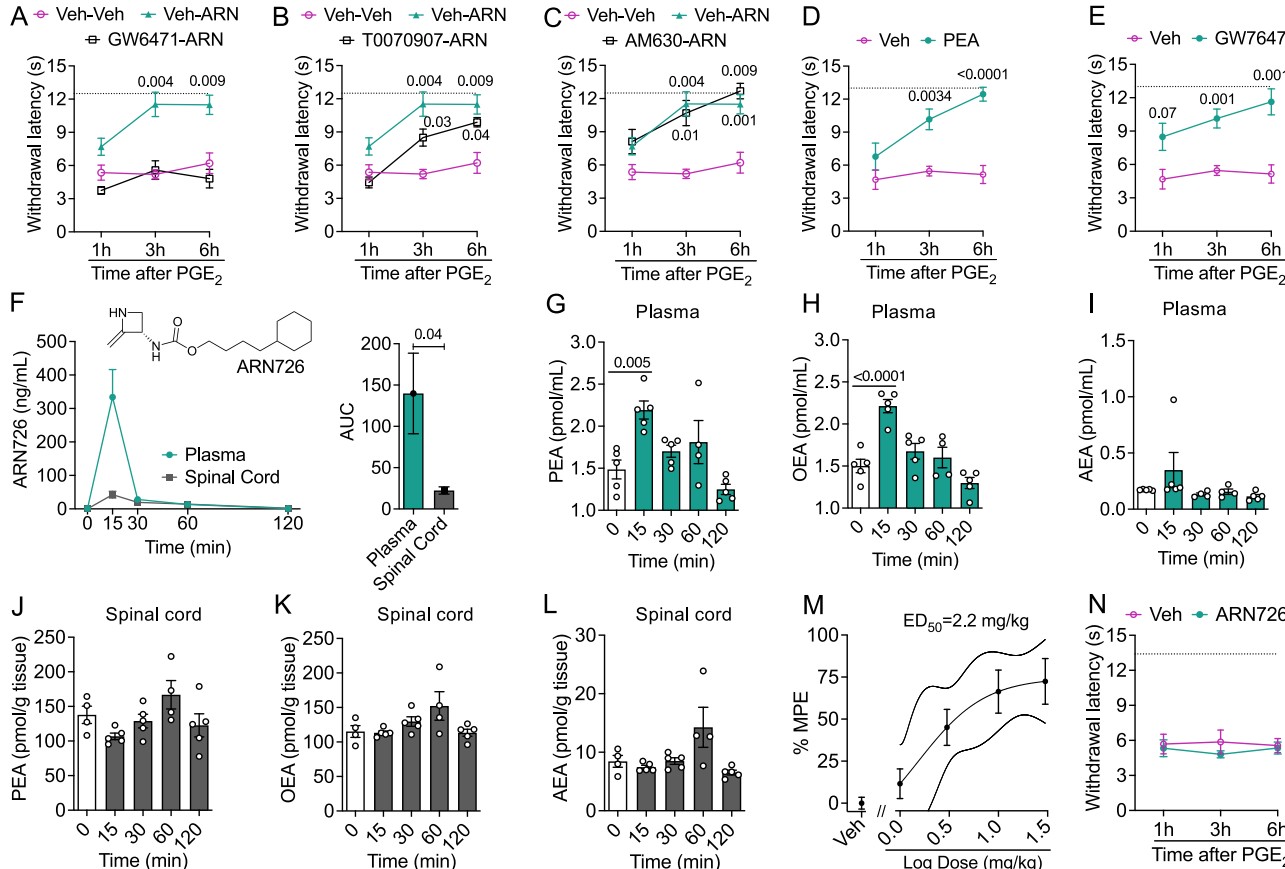

**Fig. 2 | Role of peripheral NAAA and PPAR-α signaling in initiation of hyperalgesic priming.** **A**–**C** IL-6-primed mice were treated on 1d-3d with ARN19702 (ARN, 30 mg·kg⁻¹) alone or in combination with GW6471 (4 mg·kg⁻¹; $n = 8$), T0070907 (1 mg·kg⁻¹; $n = 7$), AM630 (3 mg·kg⁻¹; $n = 8$), or an appropriate vehicle ($n = 8$). Effects of PGE₂ in mice treated with the following: (**A**) ARN19702 alone (green triangles), ARN19702 plus GW6471 (open squares), or vehicle (Veh, magenta circles); (**B**) ARN19702 alone (green triangles), ARN19702 plus T0070907 (open squares), or vehicle (magenta circles); and (**C**) ARN19702 alone (green triangles), ARN19702 plus AM630 (open squares), or vehicle (magenta circles). (**D, E**) IL-6-primed mice were treated on 1d-3d with PEA (30 mg·kg⁻¹; $n = 8$) or GW7647 (10 mg·kg⁻¹; $n = 8$). Effects of PGE₂ in mice treated with (**D**) PEA (green circles) or vehicle (magenta circles); and (**E**) GW7647 (green circles) or vehicle (magenta circles). **F** Time-course of ARN726 concentrations in plasma (green circles) and spinal cord (gray squares) after IP administration in mice (10 mg·kg⁻¹; $n = 5$ per group/time point). Top, chemical structure of ARN726. Right panel, area under the curve (AUC). **G**–**I** Effect of ARN726 (10 mg·kg⁻¹) on plasma concentrations of (**G**) PEA, (**H**) OEA, and (**I**) anandamide (AEA). Gray bars, baseline plasma concentrations. **J**–**L** Effect of ARN726 (10 mg·kg⁻¹) on the spinal cord concentrations of PEA (**J**), OEA (**K**), and AEA (**L**). Open bars, baseline analyte concentrations. **M** IL-6-primed mice ($n = 7$-8 per group) were treated on 1d-3d with ARN726 (1-30 mg·kg⁻¹) and the response to PGE₂ was assessed on 6d. % MPE, percent of maximal possible effect. **N** Effects of intrathecal vehicle (magenta circles) or ARN726 (30 ng, administered on 1d and 3d) (green circles) on the response to PGE₂ in IL-6 primed mice ($n = 7$-8 per group). Data are presented as mean ± S.E.M. and were analyzed using the two-tailed unpaired Student's $t$ test (**F**, AUC), one-way ANOVA (**G**–**L**), or two-way repeated measure ANOVA (**A**–**F**, **N**). Dunnett's or Bonferroni's post hoc test was applied as needed. $P$ values versus vehicle or baseline/time 0 are indicated. Dotted lines indicate baseline withdrawal latency. Source data are provided as a Source Data file.

fertile, and had normative nociceptive thresholds and motor coordination (Fig. S9A, B). When compared to *Naaa*^fl/fl^ littermates, which were used as controls, *Naaa*^CD11b-/-^ mice phenocopied global *Naaa* knockouts in that they exhibited an attenuated acute reaction to IL-6 (Fig. 3D) and were resistant to priming (Fig. 3E). The results suggest that NAAA-expressing CD11b⁺ myeloid cells participate in both the initial hypersensitivity evoked by IL-6 and the emergence of HP but are indispensable only for the latter.

Experiments with transgenic mice overexpressing NAAA in CD11b⁺ cells further strengthened this conclusion. Prior work has shown that resident lung macrophages in *Naaa*^CD11b+^ mice are inherently hyperactive[41], raising the possibility that NAAA overexpression in myeloid cells might promote a constitutively primed phenotype. Consistent with this prediction, the nocifensive response to PGE₂ was heightened in male *Naaa*^CD11b+^ mice that had not been previously exposed to IL-6, but not in their wild-type littermates (Fig. 3F, G). Together, the findings indicate that NAAA activity in peripheral CD11b⁺ myeloid cells is both necessary and sufficient for the initiation of HP.

### PPAR-α in CD11b⁺ myeloid cells suppresses HP initiation
To further probe the role of lipid-dependent PPAR-α signaling in the initiation of HP, we generated mice that lack the nuclear receptor in CD11b⁺ cells (Fig. 4A–C), expecting that, like mutants overexpressing NAAA in the same cells, they would also be constitutively primed. Confirming this prediction, the nocifensive response to PGE₂ was enhanced in male IL-6-naïve *Ppara*^CD11b-/-^ mice, relative to their *Ppara*^fl/fl^ littermates (Fig. 4D, E). Of note, PPAR-α deletion in CD11b⁺ cells not only sensitized mice to PGE₂ but also negated the protective effect caused by the administration of ARN19702 during HP incubation (Fig. 4F-I). The results suggest that NAAA-regulated PPAR-α signaling in peripheral CD11b⁺ myeloid cells controls the transition to HP, most likely through a cell-autonomous mechanism, and is the primary target for the protective effect of NAAA inhibitors.

### HP initiation requires circulating monocytes
Next, we used selective cell depletion strategies to identify CD11b⁺ cell populations – blood-borne monocytes, resident macrophages, or neutrophils – that contribute to HP initiation. First, we treated wild-

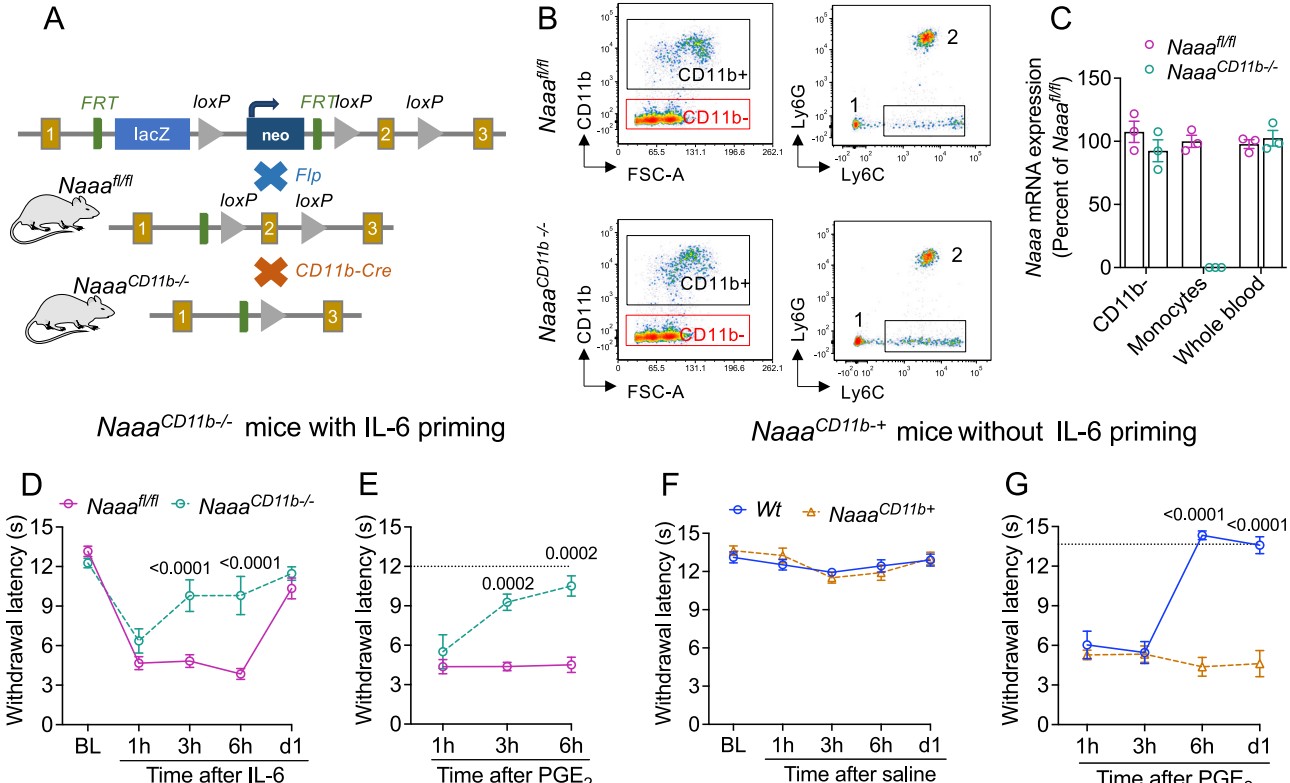

**Fig. 3 | NAAA in CD11b⁺ myeloid cells initiates hyperalgesic priming.**
**A** Generation of $Naaa^{CD11b-/-}$ mice. Cross-breeding of global $Naaa^{-/-}$ mice with flippase (fl) FLPo-10 mice produced $Naaa^{fl/fl}$ offspring, which was mated with CD11b-Cre mice to yield the $Naaa^{CD11b-/-}$ line. Yellow boxes: $Naaa$ coding sequences; green boxes, Flp recognition target (FRT) sites; gray triangles: loxP sites flanking the deleted $Naaa$ sequence. The illustration was generated in PowerPoint. Mouse image was obtained from ChemDraw. **B** FACS tracings showing isolation of monocytes from blood samples of (top panels) $Naaa^{fl/fl}$ and (bottom panels) $Naaa^{CD11b-/-}$ mice. Left, isolation of CD11b⁺ cells; right, isolation of Ly6C⁺ monocytes. The square denotes the Ly6C^high ('classical') monocyte subpopulation collected for RT-qPCR analysis. 1, Ly6C^low ('non-classical') monocytes; 2, neutrophils. **C** $Naaa$ mRNA levels (assessed by RT-qPCR) in FACS-isolated CD11b⁻ cells and Ly6C^high monocytes from $Naaa^{CD11b-/-}$ mice (green circles, $n=3$) and $Naaa^{fl/fl}$ (magenta circles,

$n=3$). Whole blood is shown for comparison. Bars represent the average of 3 biological replicates. **D** Self-resolving heat hypersensitivity elicited by IL-6 in $Naaa^{CD11b-/-}$ mice (green circles, $n=7$) and control $Naaa^{fl/fl}$ mice (magenta circles, $n=8$). **E** Effects of PGE₂ in IL-6-primed $Naaa^{CD11b-/-}$ mice (green circles) and control $Naaa^{fl/fl}$ mice (magenta circles). **F** Normal heat sensitivity in $Naaa^{CD11b+}$ mice (orange triangles, $n=8$) and their wild-type littermates ($Wt$, blue circles, $n=8$) after saline injection (20 μL). **G** Effects of PGE₂ in $Naaa^{CD11b+}$ (orange triangles) and wild-type littermates (blue circles). Data are expressed as mean ± S.E.M. and were analyzed using the two-tailed unpaired Student's $t$ test (AUC) or two-way repeated measure ANOVA. Bonferroni's *post hoc* test was applied as appropriate. $P$ values versus $Naaa^{fl/fl}$ or $Wt$ are indicated. Dotted lines indicate baseline withdrawal latency. Source data are provided as a Source Data file.

type mice with liposome-encapsulated clodronate, a peripherally restricted bisphosphonate that selectively induces apoptosis in monocyte-derived cells[47–49]. Male wild-type mice were given, with a 3-day interval, two IP injections of liposomes loaded with either clodronate or its vehicle (PBS) (Fig. 5A)[13]. FACS analyses of blood and spinal cord showed that, compared to PBS liposomes, clodronate liposomes lowered the number of circulating Ly6C^high ('classical') monocytes without affecting Ly6C^low ('nonclassical') monocytes or spinal cord microglia (Fig. 5B, C, Fig. S10A). Consistent with prior results[50], clodronate increased the number of neutrophils (Fig. 5D), likely owing to monocyte death[51]. Separate groups of liposome-treated mice were exposed to IL-6 (0.5 ng, i.pl.) and, six days later, were challenged with PGE₂ (100 ng, i.pl.) (Fig. 5E). As expected[52], administration of clodronate liposomes suppressed the initial response to IL-6 (Fig. 5F). In addition, clodronate treatment normalized the effect of PGE₂ (Fig. 5G), indicating that HP initiation was also blocked. In a second experiment, we fed male mice for three weeks a chow containing either the CSF1 receptor antagonist PLX5622 (1.2 g·kg⁻¹ chow) or its vehicle (Fig. 5H). PLX5622 treatment reduced the numbers of circulating monocytes, neutrophils (Fig. 5I–K), and spinal cord microglia (Fig. S10B)[53,54], but caused no observable change in baseline nociceptive thresholds or motor coordination (Fig. S11). Closely

matching the results obtained with clodronate, treatment with PLX5622 (Fig. 5K) prevented the acute effect of IL-6 (Fig. 5M) as well as the emergence of HP (Fig. 5N).

Lastly, we asked whether removing macrophages from the site of IL-6 administration might affect HP initiation. Two intraplantar injections of clodronate liposomes separated by a 2-day interval (Fig. S12A) caused a substantial reduction in the number of CD68⁺ macrophages in paw tissue before IL-6 administration (Fig. S12B) but no detectable effect on IL-6-induced HP (Fig. S12C). Moreover, clodronate treatment did not alter baseline nociceptive thresholds in naïve mice (Fig. S12D) but delayed the acute response to IL-6 (Fig. S12E), suggesting that local macrophages may contribute in part to such response. Together, these findings suggest that circulating monocytes (not macrophages or neutrophils) are indispensable for the initiation of HP.

## Monocytes are activated during HP incubation

To further test this hypothesis, we administered IL-6 (0.5 ng, i.pl.) or its vehicle to male mice, collected cardiac blood 72 h later (i.e., at the tail end of the incubation phase of HP), and investigated the molecular phenotype of circulating immune cells using mass cytometry by time of flight (CyTOF) (Fig. 6A). CyTOF analyses distinguished six CD45⁺ cell populations – B cells, Ly6C^high and Ly6C^low monocytes, neutrophils, NK

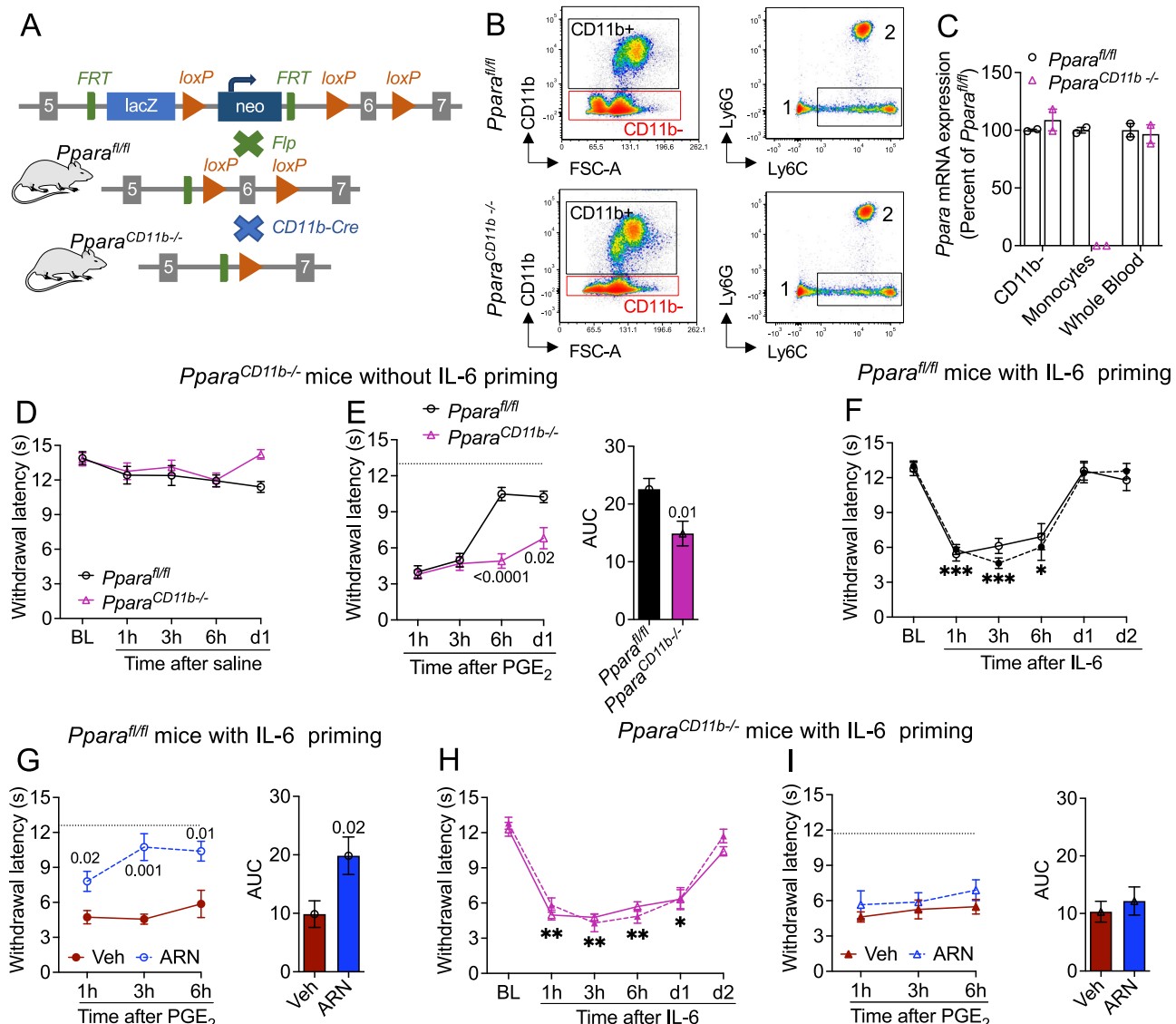

**Fig. 4 | PPAR-α in CD11b⁺ myeloid cells counters hyperalgesic priming.**
**A** Generation of *Ppara^CD11b-/-* mice. Global *Ppara^-/-* mice were mated with flippase (fl) FLPo-10 mice and the offspring was cross-bred with CD11b-Cre mice to yield the *Ppara^CD11b-/-* line. Gray boxes: *Ppara* coding sequences; green boxes, Flp recognition target (FRT) sites; gray triangles: loxP sites flanking the deleted *Ppara* sequence. The illustration was generated in PowerPoint. Mouse image was obtained from ChemDraw. **B** FACS tracings showing the isolation of monocytes from blood samples of (top) *Ppara^fl/fl* and (bottom) *Ppara^CD11b-/-* mice. Left, isolation of CD11b⁺ cells; right, isolation of Ly6C⁺ monocytes. The square denotes the Ly6C^high ('classical') monocyte subpopulation collected for RT-qPCR analysis. 1, Ly6C^low ('non-classical') monocytes; 2, neutrophils. **C** *Ppara* mRNA levels (assessed by RT-qPCR) in FACS-isolated CD11b⁻ cells and Ly6C^high monocytes from *Ppara^fl/fl* (open circles, n = 2) and *Ppara^CD11b-/-* (magenta triangles, n = 2) mice. Whole blood is shown for comparison. Bars represent the average of 2 biological replicates. **D** Normal heat sensitivity in *Ppara^fl/fl* and *Ppara^CD11b-/-* mice (n = 8 per group) after saline injection

(20 μL). **E** Effects of PGE₂ in non-primed (saline-injected) *Ppara^fl/fl* and *Ppara^CD11b-/-* mice. Right panel, area under the curve (AUC). **F, G** IL-6-primed *Ppara^fl/fl* mice were treated on 1d-3d with vehicle or ARN19702 (30 mg·kg⁻¹). **F** Effect of IL-6 before administration of vehicle (open symbols, n = 10) or ARN19702 (closed symbols, n = 10). **G** Effects of PGE₂ in IL-6-primed *Ppara^fl/fl* mice after administration of vehicle (red circles) or ARN19702 (blue circles). Right panel, AUC. **H, I** IL-6-primed *Ppara^CD11b-/-* mice (n = 7 per group) were treated on 1d-3d with vehicle or ARN19702 (30 mg·kg⁻¹). **H** Self-resolving heat hypersensitivity elicited by IL-6 before administration of vehicle (open symbols) or ARN19702 (closed symbols). (**I**) Effect of PGE₂ in IL-6-primed *Ppara^CD11b-/-* mice after administration of vehicle (red triangles) or ARN19702 (blue triangles). Right panel, AUC. Data are represented as mean ± S.E.M. and were analyzed using the two-tailed unpaired Student's *t* test (AUC) or two-way repeated measure ANOVA. ***P < 0.001, **P < 0.01, *P < 0.05, versus baseline. *P* values versus *Ppara^fl/fl* or vehicle are shown, when possible. Dotted lines indicate baseline withdrawal latency. Source data are provided as a Source Data file.

cells, and T cells (Fig. 6A) – and showed that IL-6 treatment had a profound effect on the phenotype of Ly6C^high monocytes (Fig. 6B, C). Significant increases were seen in three key markers of monocyte activation: (i) C-C chemokine receptor type 2 (CCR2), which binds the secreted chemoattractant C-C motif chemokine ligand 2 (CCL2)[55–57]; (ii) cell-adhesion protein CD43, involved in leukocyte diapedesis[58,59]; and (iii) C-X3-C motif chemokine receptor 1 (CX3CR1), which promotes monocyte migration towards sites where fractalkine is expressed[60], including dorsal root ganglia[61] (Fig. 6B). Under control conditions,

CCR2 was expressed in 44.5% of monocytes, a frequency that increased to 81.6% following IL-6 stimulation. Similarly, CD43 expression rose from 11.9% to 57.4% under the same experimental conditions (Fig. 6D). Expression of the activation-associated proteins CD64[62] and CD11c[63], as well as the ectonucleotidase CD39[64] was also heightened (Fig. 6C). By contrast, the molecular profile of Ly6C^low monocytes and neutrophils remained generally stable after Il-6 treatment (Fig. 6E, F). Noteworthy exceptions were observed, including changes in CD39 and CD43 expression in neutrophils and alterations in CD39 and CX3CR1 in

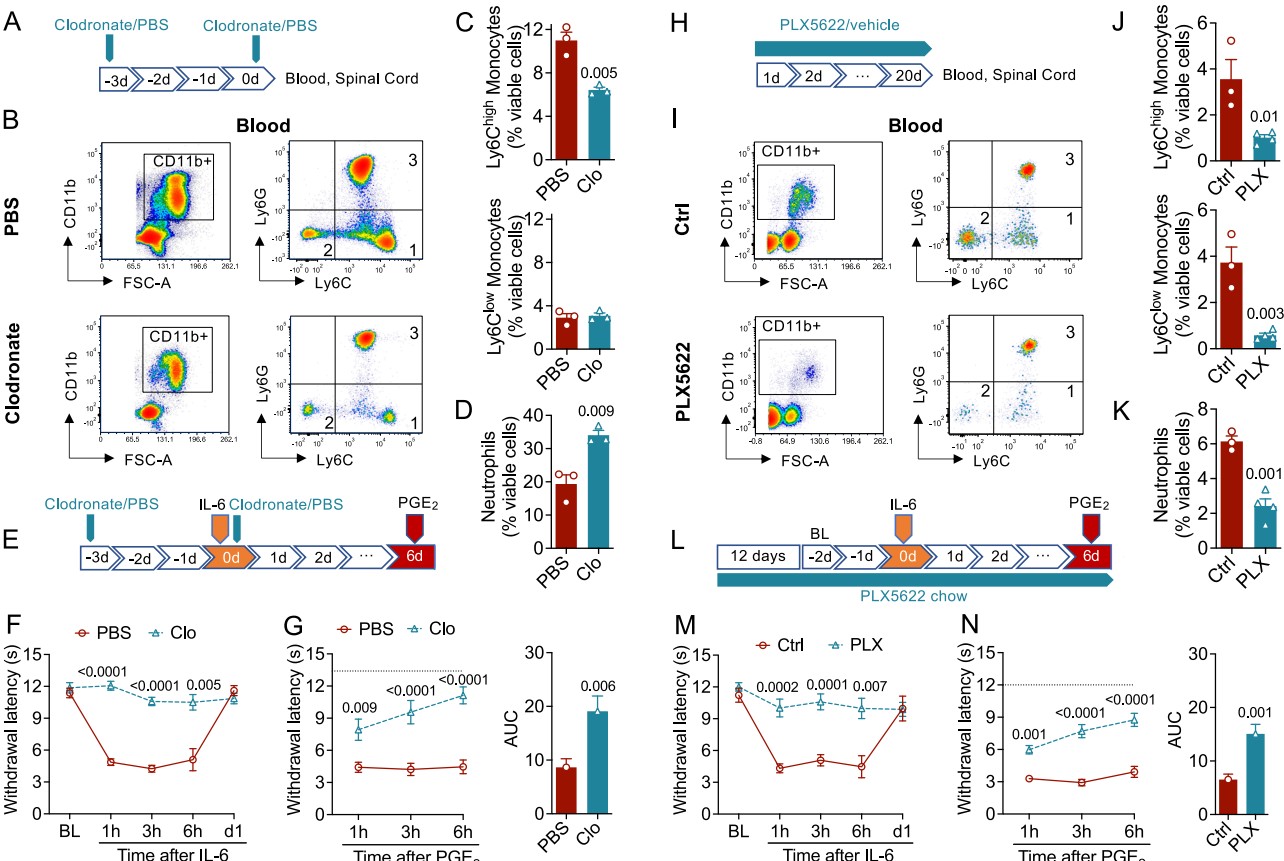

**Fig. 5 | Monocytes, not neutrophils, are required for hyperalgesic priming.**
**A−D** Verification of monocyte removal by clodronate. **A** Experimental timeline for the injection of liposomes containing clodronate or PBS. **B** FACS tracings of blood samples from mice treated with (top) PBS or (bottom) clodronate. Left, isolation of CD11b$^+$ cells; right, isolation of Ly6C$^{high}$ ('classical') monocytes (1), Ly6C$^{low}$ ('non-classical') monocytes (2), and neutrophils (3). The squares denote areas selected for quantification. **C, D** Number of circulating (**C**) monocytes and (**D**) neutrophils in mice treated with PBS (red bar, $n = 3$) or clodronate (blue bar, $n = 3$). **E** Experimental timeline. IL-6 was administered in wild-type male mice immediately before the last clodronate/PBS injection. PGE$_2$ was injected 6 days later. **F** Self-resolving heat hypersensitivity elicited by IL-6 in mice treated with PBS (red line, $n = 8$) or clodronate (Clo, blue line, $n = 8$). **G** Effects of PGE$_2$ in IL-6 primed mice treated with PBS (red line) or clodronate (Clo, blue line). Right panel, area under the curve (AUC). **H−K** Verification of monocyte removal by PLX5622. (**H**) Experimental timeline for treatment with PLX5622 or vehicle. (**I**) FACS tracings of blood samples from mice

fed a chow-containing vehicle (top) or PLX5622 (bottom). Left, isolation of CD11b$^+$ cells; right, isolation of Ly6C$^{high}$ monocytes (1), Ly6C$^{low}$ monocytes (2), and neutrophils (3). The squares denote areas selected for quantification. **J, K** Number of circulating (**J**) monocytes and (**K**) neutrophils in mice exposed to control chow (red bar, $n = 3$) and chow containing PLX5622 (blue bar, $n = 4$). **L** Experimental timeline. IL-6 was administered in wild-type male mice after 14 days of exposure to control or PLX5622-containing chow. The treatment was continued and PGE$_2$ was injected 6 days later. **M** Self-resolving heat hypersensitivity elicited by IL-6 in mice exposed to control chow (red line, $n = 8$) or PLX5622-containing chow (blue line, $n = 10$). **N** Effects of PGE$_2$ in IL-6 primed mice exposed to control chow (red line) or PLX5622-containing chow (blue line). Right panel, AUC. Data are represented as mean ± S.E.M. and were analyzed using the two-tailed unpaired Student's *t* test (AUC) or two-way repeated measure. *P* values versus PBS or control are indicated. Dotted lines indicate baseline withdrawal latency. Source data are provided as a Source Data file.

Ly6C$^{low}$ monocytes. Thus, the incubation phase of HP coincides with the emergence of one or more activated monocyte subpopulation(s) in circulation.

## Discussion

The results identify NAAA-regulated signaling at PPAR-α as a molecular switch that directs blood-borne monocytes to initiate HP following peripheral tissue damage. Four independent lines of evidence support this conclusion. First, inhibiting NAAA or activating PPAR-α during the incubation phase of HP stops its consolidation; global NAAA deletion has a similar effect. Second, mutant mice that selectively lack NAAA in CD11b$^+$ cells – which include monocytes, macrophages, and neutrophils – are resistant to priming, whereas mice that overexpress NAAA or lack PPAR-α in the same cell lineage are constitutively primed. Third, depletion of circulating monocytes, but not neutrophils or resident paw macrophages, renders mice refractory to priming. Lastly, HP incubation is accompanied by the appearance of activated monocytes in circulation. Figure 7 integrates

these findings in a hypothetical model for the role of NAAA in HP initiation.

According to this model, PGE$_2$ produces in nonprimed mice a short-lived (~2 h) nocifensive response by ligating protein kinase A-coupled EP$_2$/EP$_4$ receptors in isolectin-B4$^+$ nociceptive neurons[17,18,21] (Fig. 7, left panel). Priming agents alter this homeostatic condition in two distinct ways: on the one hand, they evoke a self-resolving inflammatory reaction whose sensory signs can be suppressed by standard analgesic drugs[22,23]; on the other, they initiate priming through an as-yet-unknown mechanism that is insensitive to such drugs (Fig. 7, center panel). The initiation phase is followed by a quiescent 72h-long incubation period that precedes stabilization of the primed state[23,24] (Fig. 7, center panel). During this time interval, NAAA activation might alter the phenotype of blood-borne monocytes (by inducing expression of CCR2, CD43, CX3CR1, etc.) thus enabling their migration to target tissues (including dorsal root ganglia) and their functional interaction with local nociceptors. Based on the present data, the neuroplastic modifications that support the maintenance of a

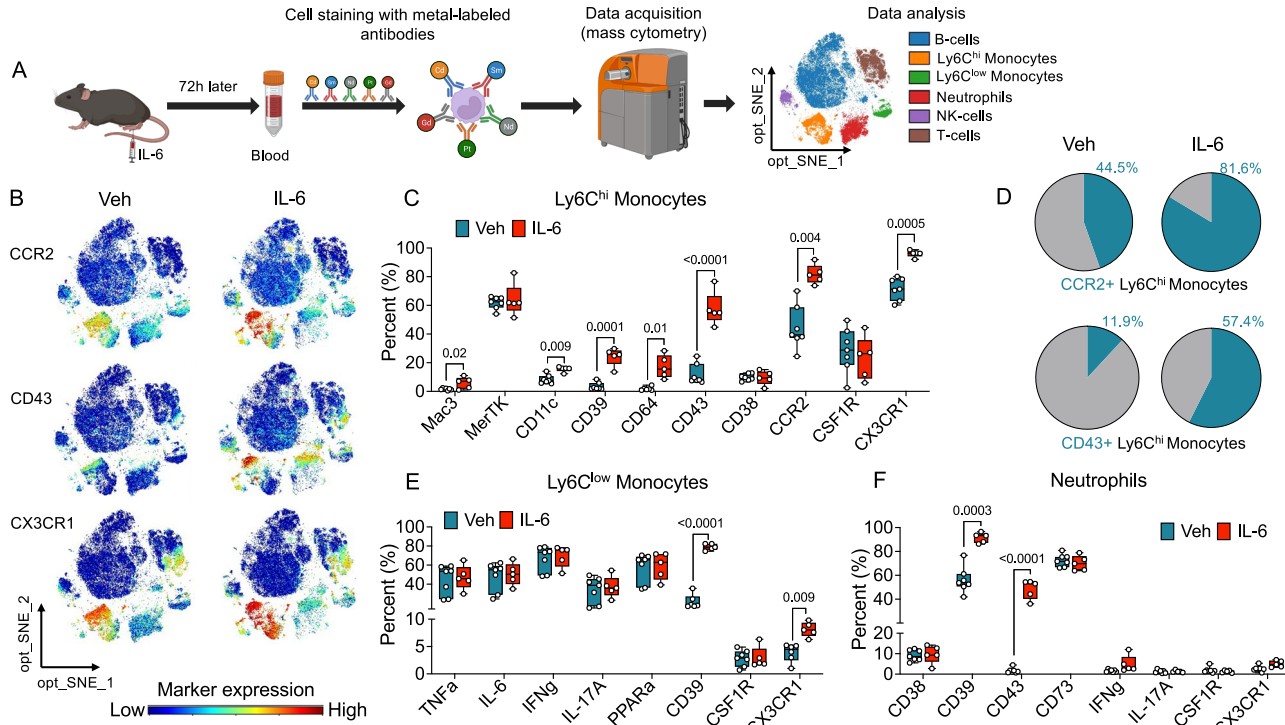

**Fig. 6 | Monocyte activation during incubation of hyperalgesic priming.**
**A** Schematic illustration of the CyTOF protocol used in these experiments. The optimized Stochastic Neighbor Embedding (opt-SNE) plot identifies six CD45+ cell populations in cardiac blood of vehicle- and IL-6-treated mice (n = 5-7 per group): B, T, and NK cells, Ly6Chigh and Ly6Clow monocytes, and neutrophils. The illustration was partially generated with BioRender.com. **B** opt-SNE plots depicting density heatmaps of cells expressing CCR2 (top), CD43 (middle), and CX3CR1 (bottom) in mice treated with vehicle (left; n = 7) or IL-6 (right; n = 5). **C** Boxplots showing quantification of activation markers in Ly6Chigh monocytes. **D** Pie charts showing

distribution of CCR2+ (top) and CD43+ (bottom) Ly6Chigh monocytes in vehicle- (left) and IL-6-treated (right) mice. Numbers indicate percentages of total cell number. **E, F** Boxplots showing quantification of activation markers in Ly6Clow monocytes (**E**) and neutrophils (**F**). Data in C, E, and F are presented in box and whiskers plots showing the median, interquartile range, the minimum and maximum values. They were obtained from 5-7 independent biological replicates and analyzed by multiple two-tailed unpaired t test with Bonferroni's correction. Adjusted P values versus vehicle are indicated. Source data are provided as a Source Data file.

primed state in these neurons – including the recruitment of protein kinase Cε-dependent transduction pathways[19–21] and the local translation and retrograde axonal transport of CREB[25] – depend on such interaction (Fig. 6, center and right panels). The model emphasizes three points: first, the role of NAAA-regulated PPAR-α signaling in controlling monocyte recruitment; second, the cooperation between activated monocytes and first-order nociceptors in the induction of priming; third, the subdivision of this process into several mechanistically distinct components. Each of these points is briefly discussed below.

The regulatory functions of PPAR-α in monocytes and macrophages have been the object of intensive investigation. Studies have shown that, when ligand-bound, PPAR-α suppresses inflammatory signaling in macrophages[65,66] and promotes their polarization toward an M2 phenotype associated with tissue protection[67,68]. There is also evidence that NAAA's main substrates, PEA and OEA[26], are the ligands responsible for these effects. First, both substances are potent PPAR-α agonists[27,29,69]. Second, monocytes and macrophages express the zinc hydrolase N-acyl phosphatidylethanolamine phospholipase D (NAPE-PLD), which constitutively generates PEA and OEA[70,71], as well as NAAA, which deactivates them[26]. Third, inflammatory stimuli suppress NAPE-PLD expression (via transcriptional regulation)[72] and enhance NAAA activity (presumably via autoproteolysis of its inactive polypeptide precursor)[73], causing an overall reduction in the amount of PEA and OEA available for signaling[30,35,74]. Fourth, NAAA inhibitors exert profound anti-inflammatory and antinociceptive effects by restoring baseline PEA and OEA levels and, consequently, enhancing PPAR-α-dependent transcription[30,34,35,75]. Finally, the incubation phase of HP is associated with the emergence of one or more monocyte

subpopulation(s) characterized by elevated expression of activation-associated proteins such as CCR2, CD43, and CX3CR1 (present study). Thus, the available evidence supports the possibility that NAAA-mediated interruption of PPAR-α signaling drives circulating monocyte into a 'primed' state that is necessary for HP induction. In this context, an important question that remains to be answered pertains to the mechanisms through which inflammatory stimuli lead to the suppression of NAPE-PLD transcription and the enhancement of NAAA activity. Elucidating such mechanisms might shed light on aspects of HP – such as the existence of sexual dimorphisms in rats[76–78] and mice[79] – which are presently unclear.

Neuroimmune interactions are critical to the establishment of persistent painful states[10,15,80]. For example, experiments in mouse models of arthritis and vincristine-induced toxicity have shown that the infiltration of blood-derived CX3CR1-expressing monocytes into the DRG drives the induction and maintenance of persistent pathological nociception[15,16]. Similarly, after nerve injury in mice, blood-borne monocytes migrate to the spinal cord, where they proliferate, differentiate, and act in synergy with microglia to initiate pain chronification[12–14]. The present study expands this body of knowledge in two ways. First, it shows that circulating monocytes, but not neutrophils or resident macrophages, are necessary for HP induction. Supporting this conclusion, we found that (i) systemically administered clodronate and PLX5622 exert opposite effects on neutrophil numbers yet both treatments deplete monocytes and prevent priming; and (ii) local administration of clodronate depletes macrophages from the priming site but has no effect on HP induction. However, the results do not exclude the possibility that neutrophils, which promote widespread persistent nociception in a mouse model of fibromyalgia[81],

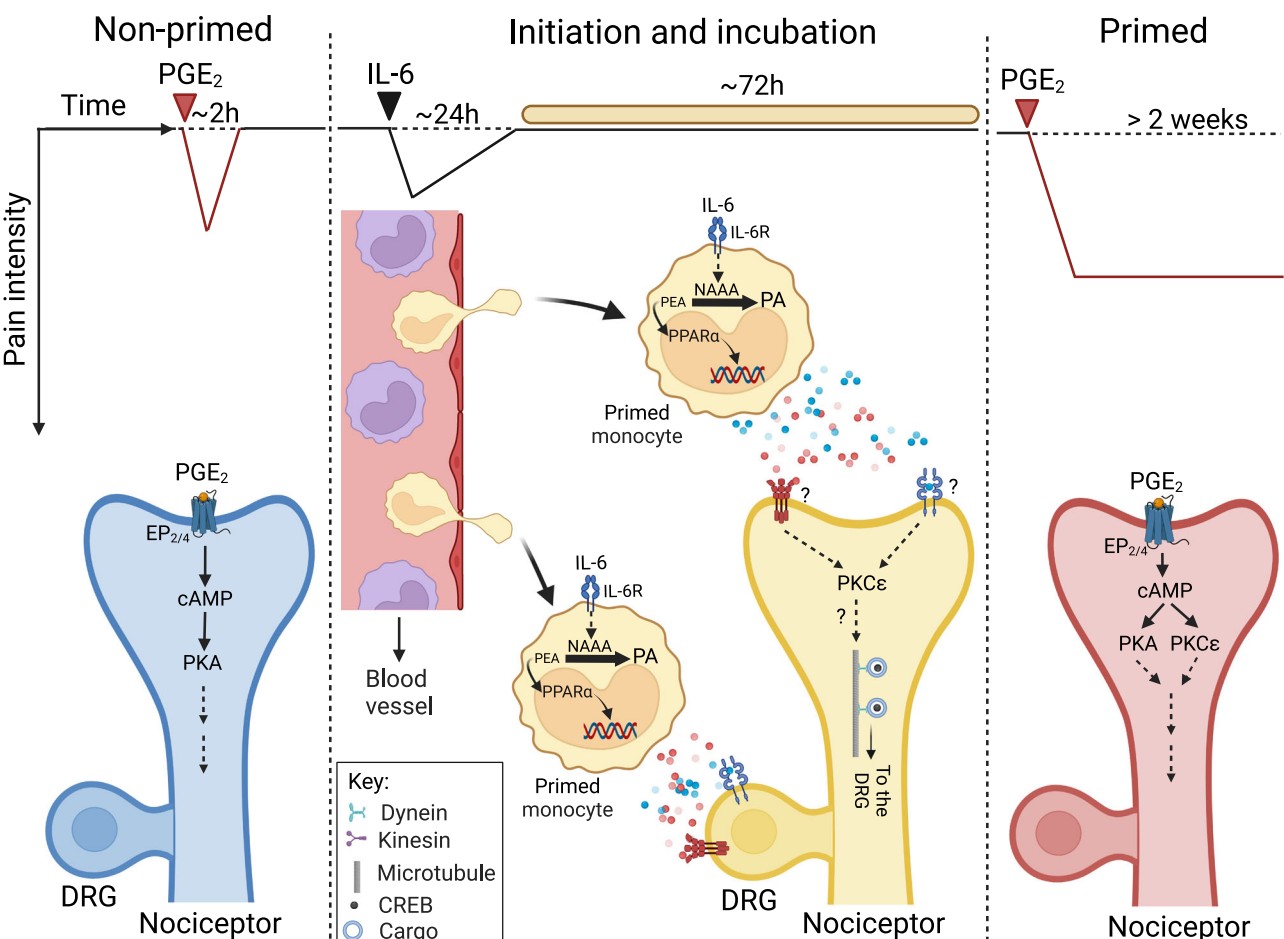

**Fig. 7 | A role for NAAA-regulated PPAR-α signaling in hyperalgesic priming.**
Left: non-primed state. In this naïve state, PGE$_2$ produces transient (~2 h) heat and
mechanical hypersensitivity by engaging protein kinase A (PKA)-coupled EP$_2$/EP$_4$
receptors in isolectin-B4$^+$ nociceptive neurons. Center: initiation and incubation.
Various cytokines (e.g., IL-6) and other noxious stimuli (e.g., growth factors, tissue
damage, etc.) evoke a self-resolving nocifensive response that lasts ~24 h and is
attenuated by standard analgesics ([22,23], present study). In parallel, the stimuli
initiate priming through a mechanism that is insensitive to analgesics. This initia-
tion phase is followed by an incubation period, which persists for ~72 h, during,
which NAAA activity interrupts PPAR-α signaling and drives monocytes into an
active (or'primed') state, which enables them to migrate to target tissues (including
the dorsal root ganglia, DRG), interact with nociceptors – presumably via one or
more unknown chemical signals (colored circles) – and effect neuroplastic changes
that initiate priming. Right: primed state. In primed animals, EP$_2$/EP$_4$ receptors in
isolectin-B4$^+$ nociceptors are coupled to an additional transduction pathway that
involves protein kinase C-epsilon (PKCε)[17]. In this state, PGE$_2$ evokes a PKCε-
mediated heat hypersensitivity that can last >2 weeks[32]. Figure created with
BioRender.com.

might play an ancillary role in HP, as also suggested by the modifica-
tions induced in this cell population by IL-6 treatment (present study).
Second, our findings identify NAAA as a molecular switch that, when
engaged, instructs monocytes to interact with first-order nociceptors
and enable priming[19–21]. The precise nature of this process is unknown
but might involve cell-cell interactions and/or the release of chemical
signals such as sphingosine-1-phosphate[82,83], reactive oxygen
species[15,84], and growth factors[85,86].

Chronic pain states are widely heterogeneous in causes, symp-
toms, impact on function, and temporal development[3]. This diversity
justifies skepticism toward a simplistic view of the progression to pain
chronicity as the transformation of one mechanistic type of pain (e.g.,
acute pain associated with injury) into another (e.g., neuropathic
pain)[87]. Nevertheless, our results do support the existence of at least
one transition point, which must be passed for lasting pathological
nociception to be established. In previous work, we found that inhi-
biting NAAA in spinal cord 48 to 72 hours after peripheral tissue
damage stops the emergence of lasting pathological nociception in
mice[31]. Similarly, in the present study, we showed that disabling per-
ipheral NAAA in the 72 hours following administration of an inflam-
matory stimulus or surgical tissue damage prevents priming

development. In both cases, the protective effects of NAAA blockade
depend on PPAR-α activation, are sexually monomorphic, and are
restricted to a three-day time window post-injury. The last property
prompted us to designate such effects with the term 'algostatic'
(ancient Greek ἄλγος 'pain' and ἱστάναι 'to stop')[31] to underscore the
fact that they are independent of the ability of NAAA inhibitors to
attenuate ongoing nociception or inflammation. The same term may
be applicable to other drugs, such as metformin, that are also able to
stop the development of persistent pathological nociception in animal
models[88–90]. Clinical studies should determine whether algostatic
agents might offer a strategy to prevent chronic pain after invasive
surgery and other kinds of physical trauma.

## Methods
### Study approval
All experimental procedures complied with the ethical regulations for
the care and use of laboratory animals promulgated by the National
Institutes of Health (NIH) and the International Association for the
Study of Pain (IASP). Formal approval was obtained from the Animal
Care and Use Committee of the University of California, Irvine (pro-
tocol # AUP-20-117).

## Chemicals

We purchased prostaglandin $E_2$ ($PGE_2$), recombinant mouse interleukin-6 (IL-6), and recombinant mouse tumor necrosis factor-alpha (TNF-$\alpha$) from R&D Systems (Minneapolis, MN). Carrageenan, gabapentin, GW7647, GW6471, T0070907, and AM630 were from Sigma-Aldrich (St. Louis, MO). Anandamide, PEA, OEA, their deuterium-containing analogs, and PLX5622 were purchased from Cayman Chemicals (Ann Arbor, MI). Levagen® PEA (a water-soluble PEA formulation) was a kind gift of GE Nutrients (Austin, TX). Liposome-encapsulated clodronate and control liposomes (containing phosphate-buffered saline, PBS, pH = 7.4) were from Liposoma (Amsterdam, Netherlands). ARN19702, ARN726, and ARN077 were prepared according to established protocols[26].

## Drug preparation and administration

Immediately before injections, a stock solution of $PGE_2$ (1 mg·mL$^{-1}$ in ethanol) was diluted in saline to a final concentration of 5 ng·$\mu$L$^{-1}$ (ethanol: 0.5% vol/vol). TNF-$\alpha$ and IL-6 were dissolved in PBS (pH = 7.4) containing fatty acid-free bovine serum albumin (BSA, 0.5%, Sigma-Aldrich) and were diluted with PBS to reach final concentrations of 5 ng·$\mu$L$^{-1}$ (TNF-$\alpha$) and 0.025 ng·$\mu$L$^{-1}$ (IL-6). ARN19702, ARN726, GW7647, GW6471, T0070907, and AM630 were dissolved in a mixture of PEG-400/Tween-80/sterile saline (15:15:70, vol/vol/vol). Carrageenan, gabapentin, and PEA were dissolved in sterile saline. Drugs and vehicles were administered by intraperitoneal (IP) or subcutaneous (SC) injection in a volume of 10 mL·kg$^{-1}$. Priming stimuli (IL-6, TNF-$\alpha$, carrageenan) and $PGE_2$ were administered into the mouse plantar surface using a hypodermic needle (30 gauge) attached to a Hamilton syringe (50 $\mu$L). Intrathecal injections of ARN726 were performed in mice anesthetized with isoflurane and held by the pelvic girdle. A Hamilton syringe (30 gauge, 10 $\mu$L) was inserted perpendicularly to the skin at the intervertebral space L5-L6 (between the hip bones). A reflexive lateral tail flick indicated a subarachnoid puncture. A volume of 5 $\mu$L was delivered and the syringe was rotated and removed before returning the mice to their home cages.

## Experimental animals

We used male and female C57BL/6 J mice (22-30 g, 10-12 weeks old) from Jackson Laboratory (Sacramento, CA) as well as the genetically modified mouse lines described below. All animals were housed in the animal facility of the University of California Irvine, using standard ventilated clear plastic cages (3-5 per cage) and conventional wood chips bedding. Food and water were available *ad libitum*. The mice were maintained in a pathogen-free environment on a 12-hr light/dark cycle at controlled temperature (22 °C) and humidity (50-60%). They were randomly assigned to experimental or control groups and behavioral testing was conducted during the light phase of the light/dark cycle. All possible efforts were made to minimize the number of animals used and their discomfort.

**Global NAAA knock-out (Naaa$^{-/-}$) mice.** Mice constitutively lacking the *Naaa* gene [(Naaa$^{-/-}$), B6N-A$^{tm1Brd}$Naaa$^{tm1a(KOMP)wtsi}$/WtsiH], were acquired from MRC Harwell (Didcot, UK) through the EMMA-European Mouse Mutants Archive. Their generation and phenotypic profile are reported here[31,35].

**CD11b$^+$ myeloid cell-specific NAAA knock-in (Naaa$^{CD11b+}$) mice.** Transgenic mice overexpressing *Naaa* in CD11b$^+$ myeloid cells (Naaa$^{CD11b+}$) were produced by GenOway (France). Their generation and phenotypic profile are reported here[31,41].

**CD11b$^+$ myeloid cell-specific NAAA knock-out (Naaa$^{CD11b-/-}$) mice.** NAAA-floxed mice (Naaa$^{fl/fl}$) were generated by mating Naaa$^{-/-}$ mice with flippase (Flp) FLPo-10 mice (Jackson Lab, #011065). Subsequent recombination of Naaa$^{fl/fl}$ with CD11b-Cre mice

(Jackson Lab, #019696)] yielded Naaa$^{CD11b-/-}$ mice, which were viable and fertile.

**CD11b$^+$ myeloid cell-specific PPAR-$\alpha$ knock-out (Ppara$^{CD11b-/-}$) mice.** PPAR-$\alpha$ floxed (Ppara$^{fl/fl}$) breeders were a kind gift of Dr Andrew S. Greenberg (Tufts University School of Medicine). Ppara$^{fl/fl}$ mice were cross-bred with CD11b-Cre animals (Jackson Lab) to obtain Ppara$^{CD11b-/-}$ mice, which were viable and fertile.

## Depletion of monocytes and other cells

C57BL6/J mice were fed *ad libitum* for 20 days either with normal chow or with chow containing the colony-stimulating factor 1 (CSF$_1$) receptor antagonist PLX5622 (1.2 g·kg$^{-1}$ of chow)[91]. Other groups of C57BL6/J mice were given systemic injections of liposomes containing PBS (control) or clodronate (16 mL·kg$^{-1}$, IP), 3 days before and immediately after IL-6 injection[13,92]. Paw-resident macrophages were deleted by administering two subcutaneous injections of liposomes (40 $\mu$L) containing clodronate (5 mg·mL$^{-1}$) or PBS into the plantar aspect of the right hind paw at a 2-day interval. Previous work has shown that this protocol removes >60% of resident macrophages[93,94].

## Fluorescence Activated Cell Sorting (FACS)

Blood was drawn via cardiac puncture and incubated with ammonium-chloride-potassium (ACK) buffer for erythrocyte lysis. Leukocytes were stained with anti-CD11b, Zombie NIR, anti-Ly6C, and anti-Ly6G antibodies (Table S1) in antibody-staining buffer (0.2% BSA and 0.1% sodium azide in PBS) for 30 min at 4 °C. Viable cell populations of interest were collected by FACS and quantified using FCS Express 7 (De Novo Software, Pasadena, CA). Gating strategies are illustrated in Figure S1. First, to separate debris from cells, we gated on the side scatter area (SSC-A) versus the forward scatter area (FSC-A) and selected the cell population with a high FSC-A/SSC-A ratio. This fraction was subjected to forward scatter height (FSC-H) versus FSC-A analysis to isolate single cells that displayed a ratio of ~1. The single-cell population was further processed to remove dead cells. Zombie NIR signal-negative cells were collected for the next steps. All FACS processes included these initial gating steps to ensure that only viable single cells were analyzed. From this population, we gated cells based on the expression of cellular marker proteins to isolate cell types of interest. To focus on innate immune cells, we selected cells that were positive for CD11b by plotting the viable cell population as CD11b versus FSC-A and selecting the population that was CD11b$^+$. This was plotted as Ly6C versus Ly6G with the Ly6C$^{high}$ cells being divided into groups depending on Ly6G expression. Monocytes were categorized as 'classical' (CD11b$^+$ Ly6C$^{high}$ Ly6G$^{low}$) or 'non-classical' (CD11b$^+$ Ly6C$^{low}$ Ly6G$^{low}$), while neutrophils were defined as CD11b$^+$ Ly6C$^{high}$ Ly6G$^{high}$ [95]. Expression of the *Naaa* and *Ppara* genes in monocytes and neutrophils was quantified by reverse transcription-quantitative polymerase chain reaction (RT-qPCR), as described below, to confirm successful recombination.

The spinal cord underwent enzymatic digestion and mechanical dissociation using a brain dissociation kit (Miltenyi Biotec, Bergisch Gladbach, Germany), following the manufacturer's instructions. Subsequently, the dissociated spinal cord cells were stained with anti-CD11b and anti-CD45 antibodies (Table S1) in antibody-staining buffer (0.2% BSA and 0.1% sodium azide in PBS) for 30 min at 4 °C. Viable cell populations were collected by FACS and quantified using FCS Express version 7.18.0025 (De Novo Software, Pasadena, CA). Gating strategies are described above. Microglia was plotted as CD45 versus CD11b, which are both established markers for this cell lineage[96].

## Mass cytometry by time of flight

Mass cytometry by time of flight (CyTOF) analyses were conducted as previously described[97]. Briefly, whole blood was lysed with 1X ACK lysis buffer, quenched with 10% serum in DMEM and centrifuged for

5 minutes at 300xg. Cell pellets were stained for viability using Cell-ID™ Cisplatin (Standard Biotools, San Francisco, CA, cat # 201064) and incubated with Fc block (Biolegend, cat # 101320) for 10 minutes on ice. Antibodies, provided by the UC Irvine Stem Cell Research Center Flow & Mass Cytometry Core's Antibody bank, were conjugated using Maxpar® X8 (Standard Biotools, cat # S201142B) and MCP9 [(cadmiums), Standard Biotools, cat # 201111 A] metal labeling kits, according to manufacturer's instructions. Antibody cocktails (100 μL, see Table S3) were prepared in Maxpar® cell staining buffer [(CSB), Standard Biotools, cat # 201068], to target both surface and intracellular/nuclear markers in a maximum of $3 \times 10^6$ cells. After staining, cells were fixed and permeabilized with Maxpar® Fix I Buffer (1 mL, cat # 201065) and Maxpar® Perm-S Buffer (2 mL, cat # 201066), respectively. Intracellular and nuclear antibodies were added to freshly washed pellets and incubated for 16 hours at 4 °C. After washing, cells were fixed with paraformaldehyde (PFA, 1 mL, 1.6%) for 10 minutes at room temperature, labeled with iridium nucleic acid (191/193Ir DNA intercalator, Standard Biotools, cat # 201192 A) and prepared for acquisition using Maxpar® Cell Acquisition Solution (CAS, Standard Biotools, cat # 201241). Samples were resuspended in a 0.1x EQ normalization bead solution (10% beads, 90% CAS), and acquisition was performed on Helios® (Standard Biotools) after tuning for optimal instrument performance with CyTOF Tuning solution (Standard Biotools, cat # 201072). CyTOF Software version 7.0.8493® was used for data analysis. Dimensionality reduction and high-dimensional data visualization were accomplished using optimized Stochastic Neighbor Embedding (opt-SNE).

### Tissue immunofluorescence

Mice were euthanized with $CO_2$ and hind paw skin was dissected and fixed for 4 hours in 4% PFA, followed by overnight cryoprotection in sucrose (30% in PBS, pH 7.4) at 4 °C. The tissues were embedded in optimal cutting temperature medium (Tissue-Tek®, Sakura Finetek, Torrance, CA) and were flash-frozen in cold isopentane. Three series of transversal sections (thickness 8 μm) were prepared using a cryostat. After rinsing with PBS, the sections were blocked and permeabilized with 0.1% Tween-20 in PBS containing 5% normal donkey serum for 30 min at room temperature. Immunostaining was performed by overnight incubation at 4 °C with an anti-CD68 primary antibody (Serotec, FA-11, 1:200, #MCA1957), followed by incubation for one hour at room temperature with a secondary anti-rat Alexa Fluor 594 donkey antibody (Invitrogen cat. #A21209; 1:200). Nuclei were stained with DAPI (1:250 in PBS) for 10 min at room temperature and slides were mounted with fluorescence-safe mounting medium. Images were obtained at 10x or 40x magnification using a Keyence BZ-X710 fluorescence microscope coupled with BZ-X Analyzer software (Keyence).

### RNA analyses

CD11b- cells and monocytes were homogenized using QiaShredder spin columns (Qiagen, Germantown, MD) and total RNA was isolated using the RNeasy Micro Kit (Qiagen). Total RNA was isolated from whole blood using the PureLink™ RNA Mini Kit (Invitrogen, Waltham, MA), following manufacturer's instructions. Before purification, samples were rendered genomic DNA free passing isolated RNA through a gDNA Eliminator spin column (Qiagen). RNA concentration and purity were determined using a SpectraMax M5 spectrophotometer (Molecular Devices, San Jose, CA). cDNA was synthesized using 20 ng of total RNA as input for the High-Capacity cDNA RT Kit with RNase Inhibitor (Applied BioSystems, Foster City, CA) with a final reaction volume of 20 μl. First-strand cDNA was amplified using TaqMan™ Universal PCR Master Mix (Invitrogen). RT-PCR primers and fluorogenic probes were purchased from Applied Biosystems [TaqMan(R) Gene Expression Assays]. We used TaqMan gene expression assays for mouse *Actb* (Mm00607939_s1), *Hprt* (Mm00446968_m1), *Gapdh*

(Mm99999915_g1), *Ppara* (Mm00440939_m1), and *Naaa* (Mm00508965_m1) (Applied Biosystems). RT-PCRs were performed in 96-well plates using CFX96 Real-Time System (Bio-Rad, Hercules, USA). The Bestkeeper software was used to determine expression stability and the geometric mean of three different housekeeping genes (*Actb, Hprt, and Gapdh*). The relative quantity of genes of interest was calculated by the $2^{-\Delta\Delta Ct}$ method and expressed as fold change over controls.

### Behavioral experiments

**HP initiation.** We produced HP by subcutaneous administration of IL-6 (0.5 ng, 20 μl), TNF-α (100 ng, 20 μl), or carrageenan (1%, 5 μL) into the plantar aspect of the right hind paw[19,32]. In some experiments, HP was induced with a surgical incision. The mice were anesthetized with 2.5% isoflurane in oxygen and a 1-cm longitudinal incision was made under aseptic conditions on the skin and fascia of the plantar surface of the right hind paw. The plantaris muscle was carefully lifted and incised longitudinally. Hemostasis was achieved by applying gentle pressure, and the skin was closed with a single 4-0 nylon suture (Ethicon, USA). After surgery, the mice were returned to their home cages. HP induction was confirmed 6 days after priming by injecting $PGE_2$ (100 ng, 20 μL) in the primed paw. $PGE_2$-induced heat hypersensitivity lasting ≥6 days indicated successful HP induction[19].

**Heat hypersensitivity.** We measured thermal hypersensitivity using a Hargreaves plantar test apparatus (San Diego Instruments, USA), as described[31,98,99]. Withdrawal latency was defined as the time (in seconds) at which mice withdrew their paws from the heat stimulus. A 15-s cut-off time was applied. Baseline latency was assessed in 2 consecutive readings made 2 days before the administration of priming agents.

**Mechanical hypersensitivity.** Mechanical hypersensitivity was assessed using a dynamic plantar aesthesiometer (Ugo Basile, Italy)[31,98,99]. Withdrawal threshold was defined as the force (in grams) at which mice withdrew their paws. A 5 g cut-off pressure was applied. Baseline threshold was assessed in 2 consecutive readings made 2 days before the administration of priming agents.

**Nociceptive thresholds.** Nociceptive thresholds were measured using the tail immersion test[100]. Briefly, mice habituated to handling were restrained in a soft tissue pocket and tail withdrawal latency was measured by dipping the distal half of the tail into a water bath set at 54 °C. Cut-off time was 10 s. Two tail-withdrawal measurements (separated by 30 s) were recorded and averaged.

**Motor coordination.** Motor coordination was evaluated using an accelerating Rotarod apparatus (Ugo Basile; rod diameter: 2 cm). Briefly, mice were familiarized to the procedure in three consecutive sessions with speed gradually increasing from 4 to 40 rpm over a 5-min period. Experiments were conducted 24 h after the last training session.

### Pharmacokinetic (PK) and pharmacodynamic experiments

Male mice (*n* = 5 per group) were given a single injection of ARN726 (10 mg-kg$^{-1}$, IP) or vehicle and were euthanized under isoflurane anesthesia 15, 30, 60, and 120 min later. Blood (0.5 mL) was collected by cardiac puncture into EDTA-rinsed syringes and transferred into polypropylene plastic tubes (1 mL) containing spray-coated K2-EDTA. Plasma was prepared by centrifugation at 1450 x *g* at 4 °C for 15 min and transferred into polypropylene tubes, which were immediately frozen and stored at −80 °C. Spinal cords were removed by hydraulic extrusion, snap-frozen on dry ice, and stored at −80 °C until processed.

**Analyte extraction.** We extracted PEA, OEA, anandamide, and ARN726 from plasma and spinal cord tissue, as described[97,99,101]. Briefly, plasma

(0.1 mL) was transferred into 8-mL glass vials, and proteins were precipitated by the addition of 0.5 mL ice-cold acetonitrile containing 1% formic acid and the following internal standards: [$^2H_4$]-PEA (20 nM), [$^2H_4$]-OEA (1 nM), [$^2H_4$]-anandamide (10 nM), and ARN077 (200 nM). Frozen spinal cords were pulverized on dry ice. Spinal tissue samples (~20 mg each) were transferred into 2 ml Precellys CK-14 soft tissue tubes (Bertin, Rockville, MD) and homogenized in 0.5 mL of ice-cold acetonitrile containing 1% formic acid and the internal standards listed above. Samples were homogenized for 1 min using a Bertin homogenizer at 4 °C in 15 s/cycle for 2 cycles with 20 s pause between cycles. Samples were stirred vigorously for 30 s and centrifuged again at 830 x $g$ at 4 °C for 15 min. The supernatants were loaded onto pre-washed [with water/acetonitrile (1:4 vol/vol)] Enhanced Matrix Removal (EMR)-Lipid cartridges (Agilent Technologies, Santa Clara, CA) and eluted under positive pressure (3-5 mmHg). Tissue pellets were rinsed with wash solution (0.2 mL), stirred for 30 s, and centrifuged at 830 x $g$ for 15 min at 4 °C. The supernatants were collected, transferred onto EMR cartridges, eluted, and pooled with the first eluate. The cartridges were washed again with wash solution (0.2 mL) and pressure was gradually increased to 10 mmHg for maximal analyte recovery. Eluates were dried under a gentle stream of $N_2$ and reconstituted in 0.1 mL methanol containing 0.1% formic acid. Samples were transferred to deactivated glass inserts (0.2 mL) placed inside amber glass vials (2 mL, Agilent Technologies).

**Lipid quantification.** PEA, OEA, and anandamide were fractionated using a 1260 series LC system (Agilent Technologies) consisting of a binary pump, degasser, thermostated autosampler, and column compartment coupled to a 6410B triple quadrupole mass spectrometric detector (MSD; Agilent Technologies). Analytes were separated on an Eclipse PAH column (1.8 μm, 2.1 × 50 mm; Agilent Technologies) with a mobile phase consisting of 0.1% formic acid in water as solvent A and 0.1% formic acid in methanol as solvent B. A linear gradient was used: 75.0% B at 0 time to 80.0% B in 5.0 min, change to 95% B at 5.01 min continuing to 6.0 min, change to 75% B at 6.01 min, and hold till 8.0 min for column re-equilibration and stop time. The column temperature was maintained at 45 °C and the autosampler temperature at 10 °C. The injection volume was 2 μL, the flow rate was 0.3 mL/min, and the total analysis time was 15.5 min. To prevent carryover, the injection needle was washed three times in the autosampler port for 30 s before each injection, using a wash solution consisting of 10% acetone in water/methanol/isopropanol/acetonitrile (1:1:1:1, vol). The MSD was operated in the positive electrospray ionization (ESI) mode. Analytes were quantified by multiple reaction monitoring (MRM) of the following transitions: anandamide = $m/z$ 348.3 > 62.2, [$^2H_4$]-anandamide = $m/z$ 352.3 > 66.2, PEA = $m/z$ 300.3 > 62.2, [$^2H_4$]-PEA = $m/z$ 304.3 > 66.2, OEA = $m/z$ 326.3 > 62.1, [$^2H_4$]-OEA = $m/z$ 330.3 > 66.1. Capillary and nozzle voltages were 3,000 and 1,900 V, respectively. The drying gas temperature was 300 °C with a flow of 5 mL/min. The sheath gas temperature was 300 °C with a flow of 12 mL/h. Nebulizer pressure was set at 40 psi. We used the MassHunter software version B.08.00 (Agilent Technologies) for instrument control, data acquisition, and analysis.

**ARN726 quantification.** ARN726 was fractionated using an Eclipse Plus $C_{18}$ column [1.8 μm, 2.1 × 30 mm; (Agilent Technologies)] with a mobile phase of 0.1% formic acid in water as solvent A and 0.1% formic acid in methanol as solvent B. A linear gradient was used: start 60% B to 80% B in 2.5 min, changed to 95% B at 2.51 min and continued till 3.0 min, changed to 60% B at 3.01 min and continued till 5.0 min. The flow rate was 0.3 mL/min till 3.0 min and then 0.4 mL/min from 3.01 min to 5.0 min. The column temperature was maintained at 40 °C and the auto-sampler temperature was 9 °C. The injection volume was 2 μL. The MS was operated in positive mode. ARN726 was quantified by MRM using ARN077 as an internal standard and the following transitions: ARN726 = $m/z$ 269.19 > 83.1, ARN077 = $m/z$ 292.16 > 74.1.

Capillary and nozzle voltages were 1800 and 1200 V, respectively. Nebulizer pressure was set at 10 psi. Drying gas temperature was 300 °C, with a flow of 12 L/min. Sheath gas temperature was 320 °C with a flow of 10 L/min.

**PK data analysis.** Results from PK experiments were analyzed using a noncompartmental model[102]. Maximal concentration ($C_{max}$) and time at which maximal concentration was reached ($T_{max}$) were determined by visual inspection of averaged data. Other PK parameters – including area under the curve ($AUC$), elimination constant ($K_{el}$), and half-life time of elimination ($t_{1/2}$) – were determined using the equations described here[102].

### Statistical analyses
Sample size for behavioral studies was predetermined by power analysis ($\alpha = 0.05$, $1-\beta = 0.8$, effect size ~35%, $n \geq 7$ per group). For molecular and biochemical studies, sample size ($n = 4$-6 animals per group) was based on prior experience with the proposed models[31,97,99]. Data analyses were performed using GraphPad Prism version 9.1 (La Jolla, CA) under blinded conditions. All data are expressed as means ± SEM. Differences between groups were assessed by either unpaired two-tailed Student's $t$ test, multiple unpaired two-tailed $t$ test with Bonferroni's correction, or analysis of variance (ANOVA) (one-way or two-way) followed by Dunnett's or Bonferroni's post hoc test unless otherwise noted. $P$ values (shown in the figures) <0.05 were considered statistically significant.

### Reporting summary
Further information on research design is available in the Nature Portfolio Reporting Summary linked to this article.

## Data availability
All data generated in this study are readily available within the paper and its supplementary files. Source data are provided with this paper.

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

## Acknowledgements

We are grateful to the anonymous Reviewers for their constructive comments, to Parwinder Singh Uppal and Vitaly Perkov for their help with behavioral studies, to Mahmoud Singer for invaluable help during CyTOF experiment, and to GE Nutrients for the kind gift of Levagen⁺ PEA. This project was funded by the National Institute on Drug Abuse grant number 1R01DA055578-01 (to D.P.) and the National Institute of Neurological Disorders and Stroke, as well as the National Center for Complementary and Integrative Health K99/R00 award #1K99AT012658-01 (to Y.F.).

## Author contributions

Conceptualization: Y.F. and D.P. Design: Y.F. and D.P. Investigation: Y.F., A.M.T., E.S., H.L., C.M.P., K.C., F.A., V.M.S., A.S.G., S.A.V., G.S., and M.M. Data analysis: Y.F., A.M.T., E.S., H.L., C.M.P., F.A., and V.M.S. Funding acquisition: D.P. and Y.F. Project administration: D.P. Supervision: D.P. Writing, original draft: D.P., Y.F., E.S., C.M.P., and V.M.S. Writing, review and editing: Y.F. and D.P.

## Competing interests

D.P. and M.M. are inventors in patents that protect ARN19702 and other NAAA inhibitors, owned by the University of California, the University of Parma, the University of Urbino, and the Fondazione Istituto Italiano di Tecnologia (no. 13/898,225, filed 20 May 2013, published 3 April 2014; no. 62/337,744, filed 17 May 2016, published 23 November 2017). D.P. and Y.F. are inventors in a patent application that protects the algostatic effects of NAAA inhibitors, filed by the University of California (no. 63/166,134, filed 25 March 2021, published 29 September 2022). The other authors declare no competing interest. No specific aspect of the manuscript was covered in patent applications.
