## [Peer Review File · Nature Communications]

REVIEWER COMMENTS

Reviewer #1 (Remarks to the Author):

N-acylethanolamine-hydrolyzing acid amidase (NAAA) is a lysosomal enzyme abundant in monocytes and macrophages. NAAA hydrolyzes N-acylethanolamines, including palmitoylethanolamide (PEA) and oleoylethanolamide (OEA). Over the years, the authors have studied the anti-inflammatory and analgesic actions of N-acylethanolamines in vivo and have developed specific NAAA inhibitors which exhibit anti-inflammatory and analgesic effects. In the present study the authors extended their interests to the involvement of NAAA in hyperalgesic priming. They used specific NAAA inhibitors and plural kinds of NAAA-deficient mice to provide the evidence that NAAA-regulated signaling at PPAR-alpha directs monocytes and macrophages to initiate hyperalgesic priming in mice exposed to an inflammatory stimulus. This is the first report clearly showing that NAAA-regulated signaling is involved in hyperalgesic priming. Thus, the work is of significance to the field and rich in originality. The work sufficiently supports the conclusions. There are no flaws in the data analysis, interpretation and conclusions. The methodology is also sound. I think that the work meets the expected standards in our field.

I just have some minor comments.

1. In Fig. 4E, the symbols for significance should be placed near the line of CD11b^{-/-}.
2. In Fig. 4F1, the asterisks should be explained in the legend.
3. In Fig. 4G1, the symbols may be asterisks.

Reviewer #2 (Remarks to the Author):

This study shows that the induction of hyperalgesic priming by intraplantar IL-6 to intraplantar PGE2 involves monocytes and macrophages. These immune cells are the source for hydrolase NAAA that acts on two lipids (PEA and OEA) which are agonists for intracellular PPAR-alpha signalling. The interruption of this pathway in monocytes and macrophages by a peripheralized NAAA inhibitor blocks the induction of IL-6-mediated hyperalgesic priming in male and female mice.

In addition, HP induction is impaired in global NAAA knockout and following silencing of NAAA in CD11b cells and by clodronate liposome and CSF1 receptor antagonist treatments.

These data suggest that monocyte/macrophages are critical for the induction of hyperalgesic priming and in such cells lipid signalling regulated by NAAA plays a mechanistic role.

This is an interesting possibility for a novel neuro-immune interaction-mediated mechanisms in the induction of persistent pain. However, the current behavioural data in transgenic mice should be paired with characterization of monocytes/macrophages in the injected paw before and after intraplantar injections of IL-6 and PGE2.

More specifically some of the following points should be addressed:

1. Do hind paw macrophages express and regulate expression of IL-6 receptor? What levels of NAAA, PEA, OEA and palmitic acid do they contain before, 6 hours and 3 days after IL-6 injections?

2. What is the level of NAAA, PEA, OEA and PA in macrophages isolated from the hind paw of NAAACD11b^{-/-} and PparaCD11b^{-/-} mice ?

3. Which phenotypes do monocytes and macrophages acquire at 6 h and 3 days after intraplantar IL-6 and after PGE2? Are they more likely to release pro-nociceptive chemicals? (Figure 6).

4. How do the authors explain IL-6 and PGE2 pro-nociceptive effects at 1 h after injections in NAAACD11b^{-/-} in figure 3?

Reviewer #3 (Remarks to the Author):

In this manuscript, the authors aim to identify molecular mechanism of acute to chronic pain transition. They concluded a critical role of NAAA on monocyte-derived cells and PPAR- α receptors in the event by using a mouse model of hyperalgesic priming. The study is carefully designed, with various experimental approaches. While a large amount of data supported most of their conclusion, I have several major concerns on the limitation of the study.

-Model of hyperalgesic priming: I greatly appreciate the model provides us an evidence that prior priming nociceptors could lead increased sensitivity to subsequent stimulation. The authors in this study demonstrated potential underlying mechanism mediated by monocyte-derived cells-associated NAAA. However, this is a very restricted model, only limited to inflammatory triggers, with specific agents and specific doses. I am sure not if the agent or the dose changed, whether the 72hour-interval is still valid, not saying if this is valid with a non-inflammatory trigger. It may mislead knowledge users by generalize the findings.

-Pain behavioral testing methods: Only heat sensitivity was assessed in the study. Whether such priming is also effective in mechanical and cold sensitivity is unknown. Whether NAAA mediates other pain modalities following the priming was not assessed either in the study.

-Skin macrophages: Two experimental approaches to deplete monocytes used in the study are indeed not ideal. Both are more effective and suitable in depleting macrophages/microglia than monocytes. The authors repeatedly indicated that observed effects are derived from monocytes/macrophages. They

excluded CNS microglia effect, but they didn't touch anything on skin macrophages, which is essential. If we believe there is a crucial peripheral contribution of NAAA in inflammatory mediator induced priming, detailed changes of macrophages in affected paw are indispensable, as they are direct players. Such data may also help to understand why it works not before or not after 72hr interval.

-Clinical implications: It is difficult for me to imagine in what clinical setting where the findings from the current study can apply. What does this 72 hr interval, not before or not after, mean for patients who come to see a physician for an acute pain?

Thus, in general, I recognize the value and the quality of the study in identifying molecular mechanism of hypersensitivity priming in a specific setting, I feel the conclusion, including the title of the study was overstated. Chronic pain is complex, I believe we need to develop precision medicine for each specific type of chronic pain.

Reviewer #4 (Remarks to the Author):

The manuscript "NAAA-regulated lipid signaling in monocyte-derived cells controls the induction of hyperalgesic priming", by Fotio et al, provides additional mechanistical information regarding a model of the transition from acute to chronic pain, hyperalgesic priming (HP). This is an interesting study, with the experiments appropriately conducted, clearly exemplifying the involvement of the immune system in the development of persistent nociceptive sensitization. The conclusions certainly add to the characterization of such phenomenon, previously shown to play a role in the persistence of pain observed in some clinical conditions. Although I do not see a reason preventing its publication, I would appreciate if the authors could clarify or include some minor comments in the final version of the manuscript:

- Experimentally, HP is defined by the potentiation (prolongation) of the mechanical hyperalgesia induced by PGE2 (>4h) in the paw when compared to a non-primed paw (<2h), indicating the hyper-responsive state triggered by the inflammation in the primary neuron. In the current study the authors evaluate the effect of PGE2 on the thermal nociceptive threshold in primed paws. What is the time course of the response produced by PGE2 on the thermal sensitivity in a normal, non-primed paw? Including this information would help the reader to visualize the increased sensitivity of the nociceptor produced by the priming stimulus, for comparison.

- Induction of HP by carrageenan is, in rats, sexually dimorphic, modulated by estrogen at the level of PKCepsilon. In mice, HP is also sexually dimorphic. However, instead of being an "all or nothing" phenomenon as in rats, this difference in mice seems to be more complex. Can the authors include a comment about these differences in the Discussion? Since the development of HP in mice of both sexes is suggested to occur at the level of NAAA, the upstream differential mechanism could be mentioned (or speculated), specifically considering that NAAA is important only during the incubation phase, when protein translation is happening.

- Why did the authors use the tail flick method to determine nociceptive thresholds if all the relevant behavior experiments were performed using the Hargreaves plantar test in the paw?

- Were the compounds injected in the paw intradermally or subcutaneously?

The information provided by this study contribute to the knowledge about the mechanisms involved in the transition from acute to chronic pain, which is of utmost importance. I look forward to receiving the authors' response and comments added to the revised version of this article.

RESPONSE TO REVIEWER COMMENTS

We are grateful to the Reviewers for their constructive and stimulating comments. We addressed them with several new experiments and substantive text edits, highlighted in *red* in this resubmission.

Reviewer #1

N-acylethanolamine-hydrolyzing acid amidase (NAAA) is a lysosomal enzyme abundant in monocytes and macrophages. NAAA hydrolyzes N-acylethanolamines, including palmitoylethanolamide (PEA) and oleoylethanolamide (OEA). Over the years, the authors have studied the anti-inflammatory and analgesic actions of N-acylethanolamines in vivo and have developed specific NAAA inhibitors which exhibit anti-inflammatory and analgesic effects. In the present study the authors extended their interests to the involvement of NAAA in hyperalgesic priming. They used specific NAAA inhibitors and plural kinds of NAAA-deficient mice to provide the evidence that NAAA-regulated signaling at PPAR-alpha directs monocytes and macrophages to initiate hyperalgesic priming in mice exposed to an inflammatory stimulus. This is the first report clearly showing that NAAA-regulated signaling is involved in hyperalgesic priming. Thus, the work is of significance to the field and rich in originality. The work sufficiently supports the conclusions. There are no flaws in the data analysis, interpretation and conclusions. The methodology is also sound. I think that the work meets the expected standards in our field. I just have some minor comments.

We are grateful for this positive evaluation of our work.

1. In Fig. 4E, the symbols for significance should be placed near the line of CD11b^{-/-}.
2. In Fig. 4F1, the asterisks should be explained in the legend.
3. In Fig. 4G1, the symbols may be asterisks.

Done.

Reviewer #2

This study shows that the induction of hyperalgesic priming by intraplantar IL-6 to intraplantar PGE2 involves monocytes and macrophages. These immune cells are the source for hydrolase NAAA that acts on two lipids (PEA and OEA) which are agonists for intracellular PPAR-alpha signalling. The interruption of this pathway in monocytes and macrophages by a peripheralized NAAA inhibitor blocks the induction of IL-6-mediated hyperalgesic priming in male and female mice. In addition, HP induction is impaired in global NAAA knockout and following silencing of NAAA in CD11b cells and by clodronate liposome and CSF1 receptor antagonist treatments.

These data suggest that monocyte/macrophages are critical for the induction of hyperalgesic priming and in such cells lipid signalling regulated by NAAA plays a mechanistic role. This is an interesting possibility for a novel neuro-immune interaction-mediated mechanisms in the induction of persistent pain. However, the current behavioural data in transgenic mice should be paired with characterization of monocytes/macrophages in the injected paw before and after

intraplantar injections of IL-6 and PGE2. More specifically some of the following points should be addressed: 1. Do hind paw macrophages express and regulate expression of IL-6 receptor? What levels of NAAA, PEA, OEA and palmitic acid do they contain before, 6 hours and 3 days after IL-6 injections? 2. What is the level of NAAA, PEA, OEA and PA in macrophages isolated from the hind paw of NAAACD11b^{-/-} and PparaCD11b^{-/-} mice?

We thank the Reviewer for this important comment, which was echoed by a similar one made by Reviewer 3. Prompted by these suggestions, we conducted a new experiment to assess whether resident macrophages contribute to the initiation of hyperalgesic priming. We removed macrophages from the hind paws of mice with two intraplantar injections of clodronate liposomes (PBS liposomes were injected as control) and examined the effects of IL-6 administration in the same paws. The results, which are reported in new **Figure S9**, show that clodronate treatment substantially reduced the number of CD68⁺ macrophages in paw tissue (**Fig. S9B**) but did not prevent IL-6-induced hyperalgesic priming (new **Fig. S9E**).

From Fig. S9. (B) Effect of PBS-containing liposomes (left) and clodronate-containing liposomes (right) on CD68⁺ cells (resident macrophages) in mouse paw skin. **(C)** Effect of PBS- or clodronate-containing liposomes on the induction of hyperalgesic priming (response to PGE₂ six days after IL-6). See Figure S9 for additional details.

By contrast, clodronate delayed the acute response to IL-6, suggesting that local macrophages contribute in part to such response.

From Fig. S9. (E) Effect of PBS- or clodronate-containing liposomes on the acute nociceptive response to IL-6. # P < 0.05 compared to PBS (n = 8). See Figure S9 for details.

Along with the data shown in **Figure 5** (effects of systemic clodronate and PLX-5622) these findings indicate that blood-borne monocytes, rather than resident macrophages, are required for priming initiation. We modified our hypothetical model (and the title of the study) to reflect these new results (please see new **Fig. 7**).

Which phenotypes do monocytes and macrophages acquire at 6 h and 3 days after intraplantar IL-6 and after PGE₂? Are they more likely to release pro-nociceptive chemicals? (Figure 6).

Based on our results showing that monocytes, not resident macrophages, are involved in priming, we addressed this question by examining the molecular phenotype of circulating immune cells by high-resolution mass cytometry by time of flight (CyTOF). Mice received intraplantar IL-6 injections and, 72 hours later (i.e., at the tail end of the incubation phase of priming), cardiac blood was collected for CyTOF analyses. The results, which are illustrated in new **Figure 6**, indicate that the incubation phase of hyperalgesic priming coincides with the emergence of one or more subpopulation(s) of activated circulating monocytes. For example:

From Fig. 6. (B) Optimized Stochastic Neighbor Embedding (opt-SNE) plots depicting density heatmaps of cells expressing the markers of monocyte activation CCR2 (top), CD43 (middle), and CX3CR1 (bottom) in vehicle- (left) and IL-6-treated (right) mice. See new Figure 6 for details.

4. How do the authors explain IL-6 and PGE₂ pro-nociceptive effects at 1 h after injections in NAAACD11b^{-/-} in figure 3?

As pointed out by the Reviewer, **Figure 3E** shows that, one h after injection, PGE₂ exerts a nociceptive effect in both control *Naaa* fl/fl and *Naaa* CD11b^{-/-} mice:

From Fig. 3. (E) Acute nociceptive effects of PGE₂ in *Naaa* CD11b^{-/-} mice (green symbols) and control *Naaa* fl/fl mice (magenta symbols) ### P < 0.001 (n = 8). See Figure 3 for details.

This effect is expected because the 1h time point measures the acute nociceptive response to PGE₂, which is generally not affected by pharmacological or genetic interventions targeting NAAA. See, for example:

From Fig. 2. (A-E) Acute nociceptive effects of PGE₂ in IL6-treated mice. Note the persistence of the effects at the 1-h time point even in mice treated with NAAA inhibitor ARN19702 (please see Fig. 2 for details). # P < 0.05, ## P < 0.01, and ### P < 0.001 compared to veh/veh (n = 10). See Figure S9 for details.

One exception is represented by global *Naaa* ko mice, which exhibited a weaker response to PGE₂ at the 1-h time point (**Fig. 3**). This could be due to NAAA deletion in as-yet-unidentified cells that are not readily accessible to the NAAA inhibitors used in our study.

Reviewer #3

In this manuscript, the authors aim to identify molecular mechanism of acute to chronic pain transition. They concluded a critical role of NAAA on monocyte-derived cells and PPAR-α receptors in the event by using a mouse model of hyperalgesic priming. The study is carefully designed, with various experimental approaches. While a large amount of data supported most of their conclusion, I have several major concerns on the limitation of the study.

Model of hyperalgesic priming: I greatly appreciate the model provides us an evidence that prior priming nociceptors could lead increased sensitivity to subsequent stimulation. The authors in this study demonstrated potential underlying mechanism mediated by monocyte-derived cells-associated NAAA. However, this is a very restricted model, only limited to inflammatory triggers, with specific agents and specific doses. I am sure not if the agent or the dose changed, whether the 72hour-interval is still valid, not saying if this is valid with a non-inflammatory trigger. It may mislead knowledge users by generalize the findings.

Hyperalgesic priming can be induced by a variety of stimuli, including proinflammatory cytokines (e.g., IL-6, TNF-α), growth factors (e.g., NGF, CSF-1), inflammatory triggers (e.g., carrageenan), and tissue damage (e.g., surgical paw incision). Its clinical relevance has been convincingly argued by various investigators (Levine, Price, etc.) who pointed out that "... the experimental framework of the hyperalgesic priming model provides important insight into clinical chronic pain because it captures the recurrent nature of some of the most common pathological pain conditions." (Kandasamy R, Price TJ. The pharmacology of nociceptor priming. *Handb Exp Pharmacol.* 2015; 227:15-37. PMID: 25846612)

Our previous submission described the results of experiments with three distinct proinflammatory stimuli – IL-6, TNF-α, and carrageenan. Nevertheless, to address the spirit of the Reviewer's concern, we have included new data using a fourth stimulus, paw incision, which has a strong nociceptive component. The results, reported in new **Figure S7**, show

that administration of a NAAA inhibitor in the first 72h after the incision fully prevents priming initiation, lending further support to the general validity of our findings.

Pain behavioral testing methods: Only heat sensitivity was assessed in the study. Whether such priming is also effective in mechanical and cold sensitivity is unknown. Whether NAAA mediates other pain modalities following the priming was not assessed either in the study.

We assessed both heat and mechanical hypersensitivity, with similar results. Representative sets of mechanical hypersensitivity data are included in new **Figure S2** and **S4**.

Skin macrophages: Two experimental approaches to deplete monocytes used in the study are indeed not ideal. Both are more effective and suitable in depleting macrophages/microglia than monocytes.

We direct the Reviewer's attention to the results reported in **Figure 5**, which show that both approaches (clodronate and PLX-5622) substantially reduced monocyte numbers in the bloodstream.

The authors repeatedly indicated that observed effects are derived from monocytes/macrophages. They excluded CNS microglia effect, but they didn't touch anything on skin macrophages, which is essential. If we believe there is a crucial peripheral contribution of NAAA in inflammatory mediator induced priming, detailed changes of macrophages in affected paw are indispensable, as they are direct players. Such data may also help to understand why it works not before or not after 72hr interval.

We thank the Reviewer for this suggestion, which we have addressed with the new experiment described in our response to Reviewer 2. Please see above.

Clinical implications: It is difficult for me to imagine in what clinical setting where the findings from the current study can apply. What does this 72 hr interval, not before or not after, mean for patients who come to see a physician for an acute pain?

This basic science study is not meant to have immediate repercussions on clinical practice. It does have, however, potential clinical significance in that it suggests that postsurgical NAAA inhibition (but not postsurgical treatment with standard analgesics) may prevent the transition to pain chronicity, a significant problem for patients who undergo thoracotomy, knee arthroplasty, mastectomy, and other invasive surgeries.

Thus, in general, I recognize the value and the quality of the study in identifying molecular mechanism of hypersensitivity priming in a specific setting, I feel the conclusion, including the title of the study was overstated. Chronic pain is complex, I believe we need to develop precision medicine for each specific type of chronic pain.

Our views about the complexity of chronic pain are very similar to the Reviewer's, as affirmed in the Discussion section of the manuscript:

“Chronic pain states are widely heterogeneous in causes, symptoms, impact on function, and temporal development³. This diversity justifies skepticism toward a simplistic view of the progression to pain chronicity as transformation of one mechanistic type of pain (e.g., acute pain associated with injury) into another (e.g., neuropathic pain)⁹⁹.”

Thus, we are surprised by the Reviewer's statement that "the conclusion, including the title of the study was overstated." The title of our manuscript (*NAAA-regulated lipid signaling in monocytes controls the induction of hyperalgesic priming*) accurately reflects the findings presented. Our conclusion is equally cautious and, we believe, fully warranted by the data: "Clinical studies should determine whether algostatic agents might offer a strategy to prevent chronic pain after invasive surgery and other kinds of physical trauma." However, when interpretation could have been ambiguous, we replaced 'chronic pain' with 'persistent pathological nociception' (see, for example, Abstract, line 38).

Reviewer #4

The manuscript "NAAA-regulated lipid signaling in monocyte-derived cells controls the induction of hyperalgesic priming", by Fotio et al, provides additional mechanistical information regarding a model of the transition from acute to chronic pain, hyperalgesic priming (HP). This is an interesting study, with the experiments appropriately conducted, clearly exemplifying the involvement of the immune system in the development of persistent nociceptive sensitization. The conclusions certainly add to the characterization of such phenomenon, previously shown to play a role in the persistence of pain observed in some clinical conditions.

We appreciate the positive comments.

Although I do not see a reason preventing its publication, I would appreciate if the authors could clarify or include some minor comments in the final version of the manuscript.

Experimentally, HP is defined by the potentiation (prolongation) of the mechanical hyperalgesia induced by PGE2 (>4h) in the paw when compared to a non-primed paw (<2h), indicating the hyper-responsive state triggered by the inflammation in the primary neuron. In the current study the authors evaluate the effect of PGE2 on the thermal nociceptive threshold in primed paws. What is the time course of the response produced by PGE2 on the thermal sensitivity in a normal, non-primed paw? Including this information would help the reader to visualize the increased sensitivity of the nociceptor produced by the priming stimulus, for comparison.

We have added a new supplemental figure (**Fig. S2**) that contains the requested information.

Induction of HP by carrageenan is, in rats, sexually dimorphic, modulated by estrogen at the level of PKCepsilon. In mice, HP is also sexually dimorphic. However, instead of being an "all or nothing" phenomenon as in rats, this difference in mice seems to be more complex. Can the authors include a comment about these differences in the Discussion? Since the development of HP in mice of both sexes is suggested to occur at the level of NAAA, the upstream differential mechanism could be mentioned (or speculated), specifically considering that NAAA is important only during the incubation phase, when protein translation is happening.

We do not know how inflammatory challenges trigger NAAA activation and whether this process might be sexually dimorphic. But we appreciate the point raised by the Reviewer and addressed its spirit by adding the following comment to the Discussion section of the manuscript (p. 17):

"In this context, an important question that remains to be answered pertains to the mechanisms through which inflammatory stimuli lead to the suppression of NAPE-PLD

transcription and the enhancement of NAAA activity. Elucidating such mechanisms might shed light on aspects of HP – such as the existence of sexual dimorphisms in rats⁸⁸⁻⁹⁰ and mice⁹¹ – which are presently unclear.”

Why did the authors use the tail flick method to determine nociceptive thresholds if all the relevant behavior experiments were performed using the Hargreaves plantar test in the paw?

We routinely use the tail-flick method to identify potential alterations in baseline nociceptive threshold in newly developed mouse lines such as those introduced in the present study. Here, we combined tail-flick (**Fig. S9**) and Hargreaves plantar tests (**Fig. 3D** and **Fig. 4G1**) to obtain a more complete view of the lines' nociceptive phenotype.

Were the compounds injected in the paw intradermally or subcutaneously?

Compounds were injected subcutaneously, between skin and muscular fascia/tendon, as customary in this model. We clarified this point in the Materials and Methods section of the manuscript.

The information provided by this study contribute to the knowledge about the mechanisms involved in the transition from acute to chronic pain, which is of utmost importance. I look forward to receiving the authors' response and comments added to the revised version of this article.

Thank you.

REVIEWERS' COMMENTS

Reviewer #1 (Remarks to the Author):

The authors appropriately revised the manuscript in response to my comments.

Reviewer #2 (Remarks to the Author):

My points have been partially addressed.

Specifically, whilst I suggested to characterise monocytes/macrophages at the site of IL-6 injection (hind paw) the authors have looked at blood monocytes. They provide evidence that intraplantar injection of IL-6 is associated with more CCR2+ monocytes circulating in blood. However, classical monocytes will have to infiltrate tissue to come in the vicinity of primary afferent terminals and promote nociceptive signalling.

Whether CCR2+ monocytes infiltrate the site of injection remains to be established and Figure 7 schematic should indicate that this study does not provide direct evidence that “monocytes migrate to target tissue”.

New Figure 6: How do the authors explain presence of F4/80 positive cells (macrophages) in blood?

Discussion: It is possible that CCL2 is upregulated in endothelial cells, and sensory neurons and accumulates in primary afferent terminals. However, the authors provide no direct evidence that this happens in their experimental conditions.

Other points. Ref 12 line 57: This is an early observation and more recent evidence indicates no infiltration of monocytes in the spinal cord following peripheral nerve injury. The authors could rephrase their statement.

Line 167: replace “deleted” with “depleted”.

Reviewer #3 (Remarks to the Author):

The authors addressed the reviewer's concerns. I don't have further comments.

Reviewer #4 (Remarks to the Author):

I am satisfied with the revision provided by the authors. I recommend this study for publication.

Specifically, whilst I suggested to characterise monocytes/macrophages at the site of IL-6 injection (hind paw) the authors have looked at blood monocytes. They provide evidence that intraplantar injection of IL-6 is associated with more CCR2+ monocytes circulating in blood. However, classical monocytes will have to infiltrate tissue to come in the vicinity of primary afferent terminals and promote nociceptive signalling. Whether CCR2+ monocytes infiltrate the site of injection remains to be established and Figure 7 schematic should indicate that this study does not provide direct evidence that “monocytes migrate to target tissue”.

We clarified that Figure 7 provides a hypothetical model which summarizes the data presented in our study and suggests a roadmap for future experiments. Please see Figure 7 legend.

New Figure 6: How do the authors explain presence of F4/80 positive cells (macrophages) in blood?

Though unlikely, it is possible that the antibody we used cross-reacted with a different monocyte antigen. Owing to this uncertainty, we removed F4/80 data from Figure 6C.

Discussion: It is possible that CCL2 is upregulated in endothelial cells, and sensory neurons and accumulates in primary afferent terminals. However, the authors provide no direct evidence that this happens in their experimental conditions.

There is substantial evidence that peripheral nerve injury upregulates CCL2 in DRG neurons, but future experiments will have to address this question as it pertains to the model used here.

Other points. Ref 12 line 57: This is an early observation and more recent evidence indicates no infiltration of monocytes in the spinal cord following peripheral nerve injury. The authors could rephrase their statement.

Evidence indicates that spinal nerve resection does not stimulate monocytes migration toward the spinal cord (PMID 27373153, presumably the study the reviewer had in mind). The hyperalgesic priming model we used, however, is quite different and, possibly, mechanistically closer to the one used in Ref 12 (partial sciatic nerve ligation). Indeed, both hyperalgesic priming and sciatic nerve ligation exhibit a rather strong inflammatory component that is not observed after spinal nerve resection.

Line 167: replace “deleted” with “depleted”.

Done.